

# Toward improved parameterization of a macro-scale hydrologic model in a discontinuous permafrost boreal forest ecosystem

Abraham Endalamaw[1, *], W. Robert Bolton[1], Jessica M. Young-Robertson[1], Don Morton[2], Larry Hinzman[3], Bart Nijssen[4]

[1]International Arctic Research Center, University of Alaska Fairbanks, Fairbanks, Alaska, USA

[2]Boreal Scientific Computing LLC, Fairbanks, Alaska, USA

[3]University of Alaska Fairbanks, Fairbanks, Alaska, USA

[4]University of Washington, Seattle, Washington, USA

*Correspondence to*: Abraham Endalamaw (amendalamaw@alaska.edu)

**ABSTRACT**

Modeling hydrological processes in the Alaskan sub-arctic is challenging because of the extreme spatial heterogeneity in soil properties and vegetation communities. However, modeling and predicting hydrological processes is critical in this region due to its vulnerability to the effects of climate change. Coarse spatial resolution datasets used in land surface modeling poised a new challenge in simulating the spatially distributed and basin integrated processes since these datasets do not adequately represent



the small-scale hydrologic, thermal and ecological heterogeneity. The goal of this study is to improve

the prediction capacity of meso-scale to large-scale hydrological models by introducing a small-scale

parameterization scheme, which better represents the spatial heterogeneity of soil properties and

vegetation cover in the Alaskan sub-arctic. The small-scale parameterization schemes are derived from

observations and fine resolution landscape modeling in the two contrasting sub-basins of the Caribou

Poker Creek Research Watershed (CPCRW) in Interior Alaska: one nearly permafrost-free (LowP) and

one that is permafrost-dominated (HighP). The fine resolution landscape model used in the small-scale

parameterization scheme is derived from the watershed topography. We found that observed soil

thermal and hydraulic properties — including the distribution of permafrost and vegetation cover

heterogeneity — are better represented in the fine resolution landscape model than the coarse resolution

datasets. Parameters derived from coarse resolution dataset and from the fine resolution landscape

model are implemented into the Variable Infiltration Capacity (VIC) meso-scale hydrological model to

simulate runoff, evapotranspiration (ET) and soil moisture in the two sub-basins of the CPCRW.

Simulated hydrographs based on the small-scale parameterization capture most of the peak and low

flows with similar accuracy in both sub-basins compared to the parameterization based on coarse

resolution dataset. On average, small-scale parameterization improves the total runoff simulation

approximately by up to 50% in the LowP sub-basin and 10% in the HighP sub-basin from the large-

scale parameterization. This study shows that the proposed small-scale landscape model can be used to

improve the performance of meso-scale hydrological models in the Alaskan sub-arctic watersheds.

***Keywords:*** *Interior Alaska, boreal forest, hydrological modeling, scaling, parameterization, vegetation,*

*Caribou Poker Creek Research Watershed (CPCRW), Variable Infiltration Capacity (VIC)*



## 1. INTRODUCTION

The sub-Arctic region of Interior Alaska lies in the transition zone between the warm temperate region to the south and the cold arctic region to the north. This region is underlain by discontinuous permafrost and is very sensitive to climate warming [*Hinzman et al.*, 2006]. Along with climate and geology, topography is a significant factor that controls the distribution of permafrost in Interior Alaska as it has a strong control on the amount and intensity of solar radiation received at the land surface [*Morrissey and Strong*, 1986; *Viereck et al.*, 1983]. A difference in solar radiation between north and south-facing slopes supports the existence of permafrost on north-facing slopes and valley bottoms, but not on south-facing slopes [*Hinzman et al.*, 2006; *Slaughter and Kane*, 1979; *Slaughter et al.*, 1983]. Subsequently, permafrost-underlain and permafrost-free areas in this region display contrasting watershed characteristics and hydrological responses. The presence or absence of permafrost is the primary factor that creates a complex landscape mosaic of sharp spatial boundaries of contrasting vegetation cover and soil hydraulic and thermal properties, moisture dynamics, and water pathways [*Bolton*, 2006; *Bolton et al.*, 2000; *Hinzman et al.*, 2002; *Jones and Rinehart*, 2010; *Petrone et al.*, 2006; *White et al.*, 2008]. Due to these small-scale complexities associated with permafrost distribution, simulation of large-scale hydrological processes remains a challenge in the Interior Alaskan boreal forest.

The hydraulic conductivity of permafrost-affected soils is several orders of magnitude less than that of the overlying organic layers and the nearby permafrost-free soil [*Burt and Williams*, 1976; *Kane and Stein*, 1983; *Ping et al.*, 2005; *Rieger et al.*, 1972; *Woo*, 1986; *Zhang et al.*, 2009]. The ice-rich permafrost is an impermeable layer at the permafrost surface that limits the hydraulic flow to the active



layer – the thin, seasonally thawed soil layer above permafrost [*Romanovsky and Osterkamp*, 1995; *Romanovsky et al.*, 2003]. Streamflow in permafrost-dominated watersheds has been described as "flashy", responding rapidly to precipitation and snowmelt with storm hydrographs displaying a sharp rise and prolonged recession [*Bolton et al.*, 2000; *Quinton and Carey*, 2008] and relatively low

baseflow between precipitation events [*Bolton*, 2006; *Hinzman et al.*, 2002; *Kane*, 1980; *Kane and Stein*, 1983; *Kane et al.*, 1981; *Petrone et al.*, 2006; *Petrone et al.*, 2007; *Slaughter et al.*, 1983]. On the other hand, in watersheds with no or relatively low areal permafrost extent, the soil hydraulic conductivity and infiltration capacity are much higher, resulting in a slower streamflow response to precipitation and snowmelt, relatively higher baseflow between storm events, and greater residence time

of water in catchments [*Bolton et al.*, 2000; *Carey and Woo*, 2001].

In the Alaskan sub-arctic, vegetation type, density, and physiological and structural properties such as Leaf Area Index (LAI) and stomatal conductance display a strong variation between permafrost-dominated and permafrost-free soils [*Viereck and Van Cleve*, 1984; *Viereck et al.*, 1983]. These variations lead to a significant variation in the partitioning of precipitation and snowmelt into runoff and

evapotranspiration (ET), and change in soil water content between permafrost-dominated and permafrost-free soils[*Bolton et al.*, 2005; *Cable et al.*, 2014; *Hinzman et al.*, 2002; *Hinzman et al.*, 2006; *Young-Robertson et al.*, 2016]. Permafrost-affected soils typically support coniferous vegetation that is shallowly rooted, tolerant of cold and wet soils, and able to survive a short growing season [*Molders*, 2011; *Morrissey and Strong*, 1986; *Viereck and Van Cleve*, 1984; *Viereck et al.*, 1983]. In contrast, the

well-drained, relatively warm permafrost-free soils support deciduous vegetation that has higher LAI and stomatal conductance, deeper root network and greater trunk height [*Cable et al.*, 2014; *Young-*

*Robertson et al.*, 2016]. This difference in vegetation between permafrost-dominated and permafrost-free soil can further influence streamflow responses [*Naito and Cairns*, 2011] due to the large differences in the rates of and controls on ET, particularly transpiration [*Baldocchi et al.*, 2000; *Ewers et al.*, 2005] and vegetation water storage [*Young-Robertson et al.*, 2016]. In Interior Alaska, deciduous trees have higher transpiration rates and vegetation water storage compared to coniferous trees [*Cable et al.*, 2014; *Ewers et al.*, 2005; *Young-Robertson et al.*, 2016; *Yuan et al.*, 2009], which limits the availability of water for runoff in the permafrost-free watersheds.

Plot-scale and hill-slope studies have documented the differences in and relationships between soil thermal and hydrologic properties, and ecosystem vegetation composition in high latitude cold regions [*Dingman*, 1973; *Kane et al.*, 1981; *Kirkby*, 1978; *Woo*, 1976; *Woo and Steer*, 1983]. However, larger scale land-surface parameterizations and the data products used in land-surface, hydrological, and climate models do not adequately represent the complex sub-Arctic watersheds with significant spatial variability in soil and vegetation dynamics. Hydrological modeling using these coarse resolution datasets cannot produce accurate estimates of the spatially variable and basin-integrated watershed responses. Until the small-scale hydrologic processes, soil properties, and vegetation distributions are well represented, accurate large-scale hydrologic simulation and modeling remains extremely challenging [*Walsh et al.*, 2005].

The primary objective of this study is to improve the prediction capability of hydrological models in Interior Alaska's boreal forest by implementing a small-scale parameterization scheme, which represents the spatial heterogeneity of soil properties and vegetation cover, into a meso-scale

hydrological model. The study is conducted in two small sub-basins of the CPCRW (LowP and HighP sub-basins of areas 5.2 km$^2$ and 5.7 km$^2$, respectively) that are representative of the region. This study transfers the plot and hill-slope scale knowledge into a meso-scale distributed hydrological model (the VIC model) so that its application can be extended to large basins in the region. A fine resolution

landscape model – on which the small-scale parameterization scheme is based – is derived from a high resolution Digital Elevation Model (DEM). Streamflow, ET and soil moisture simulations, based on the small-scale parameterization scheme and the coarse resolution datasets, are presented and investigated in both sub-basins.

## 2.  METHODS

**2.1 Study Area**

Located approximately 50 km northwest of Fairbanks, Alaska, the Caribou Poker Creek Research Watershed (CPCRW) (centred on 65$^o$10'N and 147$^o$30'W, basin area ~101 km$^2$) is within the zone of discontinuous permafrost and in the North America boreal forest region (Figure 1). CPCRW is also within the Yukon-Tanana Uplands of the Northern Plateaus Physiographic Province [*Wahrhaftig*, 1965]

with elevations ranging from 187 to 834 m.  CPCRW has a long-term record of ecological, meteorological, and hydrological data [*Bolton*, 2006; *Bolton et al.*, 2000; *Hinzman et al.*, 2002; *Hinzman et al.*, 2003; *Knudson and Hinzman*, 2000]. The climate of CPCRW is characterized as continental with large diurnal and annual temperature variations, low annual precipitation, low cloud cover, and low humidity [*Haugen et al.*, 1982]. The average annual air temperature range at CPCRW is

very large (July mean temperature = 14.7 °C, January mean temperature = -18.4 °C) with a mean annual

temperature of -2.1 °C [Figure 2, Table 1]. The mean annual precipitation is 417 mm, of which 2/3 occurs as rainfall [*Bolton*, 2006].

Seven soil series have been identified in CPCRW [*Rieger et al.*, 1972]. We group the seven series into two general categories: permafrost-dominated soils that are poorly-drained with a thick organic layer, and permafrost-free soils that are well-drained with a shallow organic layer [*Rieger et al.*, 1972]. Permafrost in CPCRW is generally found along north-facing slopes and valley bottoms where the solar input is very low [*Haugen et al.*, 1982; *Rieger et al.*, 1972]. Soils free of permafrost are generally found on south-to southwest-facing slopes.

The vegetation distribution in CPCRW displays a strong relationship with permafrost distribution. Coniferous vegetation that consists primarily of black spruce (*Picea mariana*) is generally found in areas underlain by permafrost. Feather moss (*Hylocomium spp.*), tussock tundra (*Carex aquatilis*), and sphagnum mosses (*Sphagnum sp.*) are also found along the valley bottoms. Deciduous vegetation consists primarily of aspen (*Populus tremuloides*), birch (*Betula papyrifera*), alder (*Alnus crispa*), and sporadic patches of white spruce (*Picea glauca*), and is found on the well-drained, south-facing permafrost-free soils [*Haugen et al.*, 1982].





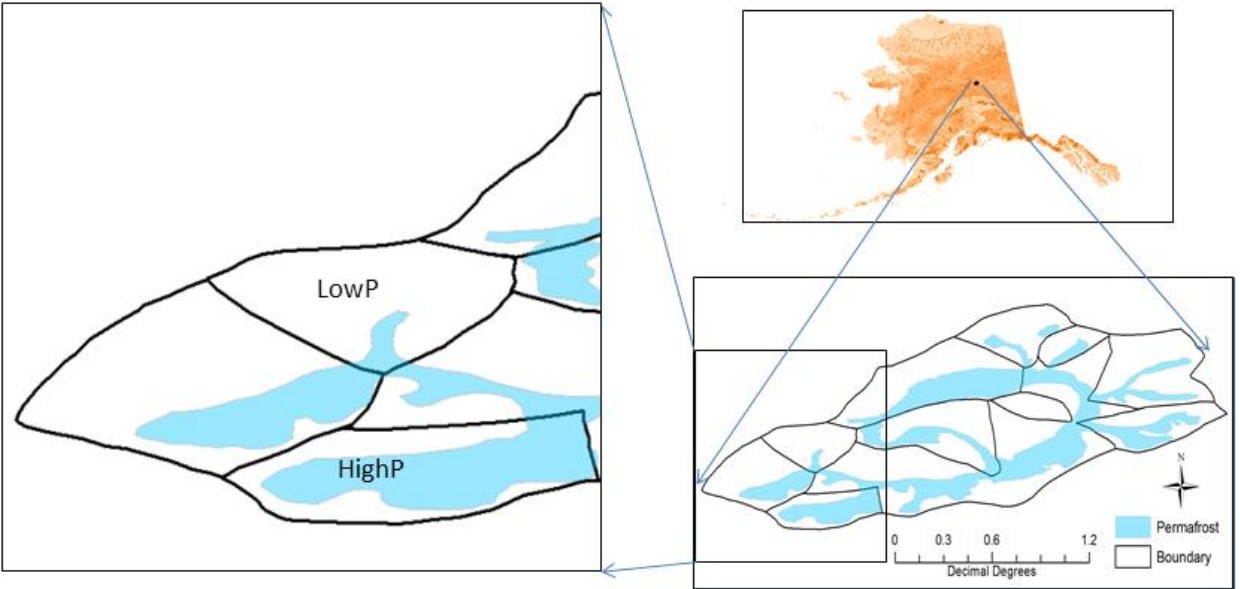

**Figure 1** Location and permafrost extent the Caribou Poker Creek Research Watershed, Alaska, and the two sub-basins of interest (low permafrost or LowP sub-basin and high permafrost or HighP sub-basin).

There are several sub-basins in the CPCRW that differ in permafrost coverage and vegetation

5  composition. The nearly permafrost-free or LowP (5.2 km$^2$) and permafrost-dominated or HighP (5.7

km$^2$) sub-basins (Figure 1) are selected for this study for their contrasting permafrost extents and

vegetation cover compositions (Table 3). Unlike vegetation cover,  permafrost extent, and solar input,

local climate does not vary between the two sub-basins [Figure 2, Table 1] due to the local understory

convective mixing of the bulk atmosphere [*Hinzman et al.*, 2006]. The coniferous/deciduous vegetation

10  composition, derived from [*Haugen et al.*, 1982], is approximately 30/70% for the LowP sub-basin and

95/5% for the HighP sub-basin (Figure 3c , Table 3). The permafrost coverage is different between the

two sub-basins, with approximately 2% and 55% in LowP and HighP sub-basins (Table 1), respectively

[*Rieger et al.*, 1972; *Yoshikawa et al.*, 2002].



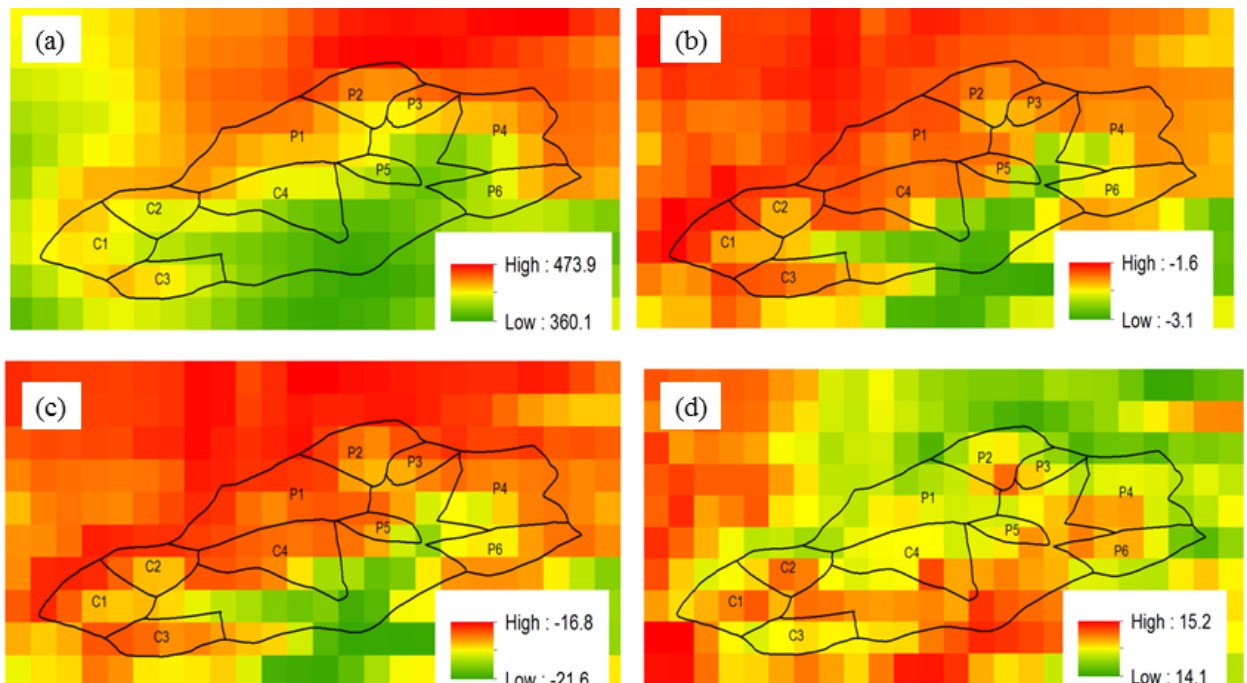

**Figure 2** Climatology of the Caribou Poker Creek Research Watershed, Alaska, from 1970 to 2012: (a) mean annual precipitation (mm), (b) mean annual air temperature ($^0$C), (c) mean January air temperature ($^0$C), and (d) mean July air temperature ($^0$C).

**Table 1** Mean (1970-2012) climatology of the Caribou Poker Creek Research Watershed (CPCRW), Alaska, and the sub-basins (LowP, HighP): MAP (mean annual precipitation, mm), MAT (mean annual air temperature, °C), MIN (mean minimum air temperature, °C), MAX (mean maximum air temperature, °C), MJan (mean January air temperature, °C), and MJuly (mean July air temperature, °C).

| basin/ Sub-basin | Permafrost extent (%) | MAP | MAT | MIN | MAX | MJan | MJuly |
|---|---|---|---|---|---|---|---|
| CPCRW | 30 | 416.7 | -2.1 | -7.2 | 3.0 | -18.4 | 14.7 |
| LowP | 2 | 421.5 | -1.9 | -6.7 | 2.9 | -17.8 | 14.7 |
| HighP | 55 | 408.2 | -2.1 | -7.3 | 3.0 | -18.5 | 14.8 |

## 2.2 Fine resolution landscape modeling

The fine resolution land scape model is a simple model used to produce high resolution soil property and vegetation cover maps for accurate representation of the watershed characteristics into the

hydrological model. Fine resolution Digital Elevation model (DEM) of the landscape is the primary input to the model. In addition, the relationship between vegetation, permafrost, slope and aspect [*Hinzman et al.*, 2006; *Morrissey and Strong*, 1986; *Viereck et al.*, 1983] is also introduced during the modeling activity.

First, aspect of the landscape is calculated using the 30m DEM [*Aster*, 2009] of the watershed with

ArcGIS aspect spatial analyst toolbox. The aspect toolbox uses elevation values of eight surrounding gird cells to calculate the gradient and aspect of each grid cell in the model domain [*Burrough et al.*, 1998] resulting in an aspect map with nine classes (Figure 3a) ) that are aggregated into two aspect classes: permafrost-underlain and permafrost-free aspects.  The following assumptions are made in the classification of each aspect into permafrost-dominated and permafrost-free aspects:

1.   North-facing slopes and valley bottoms are underlain by permafrost, and south-facing slopes are permafrost-free [*Hinzman et al.*, 2002; *Rieger et al.*, 1972; *Yoshikawa et al.*, 2002];

2.   Coniferous vegetation, primarily black spruce trees, are found along poorly-drained north-facing slopes and valley bottoms (permafrost-underlain soil) [*Haugen et al.*, 1982];



3. Deciduous vegetation, primarily birch and aspen trees, are found on the well-drained, south-facing soils (permafrost-free soil) [*Haugen et al.*, 1982]; and

4. The hydraulic conductivity of frozen soil is two-orders of magnitude less than the same soil in unfrozen conditions [*Burt and Williams*, 1976; *Kane and Stein*, 1983; *Woo*, 1986].

The small-scale observed [*Rieger et al.*, 1972] and modelled [*Yoshikawa et al.*, 2002]permafrost maps are used as a reference during the grouping of the nine aspect classes into permafrost-underlain and permafrost-free aspects. Finally, north, northeast, northwest, southwest, and flat aspects are classified as permafrost-underlain aspects with coniferous vegetation cover. South, southeast, east, and west are classified as permafrost-free aspects and with a deciduous vegetation cover. The vegetation cover and soil hydraulic property maps obtained from the fine resolution landscape model are shown in Figure 3b and Figure 4c respectively. Since each grid cell has a unique soil property, a 0.5 grid cell fraction threshold is assumed to classify a grid cell as permafrost-underlain or permafrost-free soil. We used the small-scale observed [*Rieger et al.*, 1972] permafrost map in the determination of this threshold value. As a result, a grid cell assigned with permafrost soil property when the fraction of permafrost is greater than 0.5 (Figure 4c).

### 2.3 VIC Model Description

The Variable Infiltration Capacity (VIC) model [*Liang et al.*, 1996; *Liang et al.*, 1994; *Nijssen et al.*, 1997] is a meso-scale process-based distributed hydrological model that represents vegetation heterogeneity, multiple soil layers, variable infiltration, and non-linear base flow. The version of the VIC model used in this study, VIC 4.1.2, contains several explicit formulations for snow accumulation

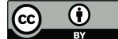

and ablation, and frozen soil, and evapotranspiration and solves the full energy balance or water balance at a sub-daily or daily time steps to simulate the energy and water fluxes for individual grid cells. A separate routing model [*Duband et al.*, 1993; *Lohmann et al.*, 1996; *Lohmann et al.*, 1998a; b] is used to collect the runoff and baseflow simulations from each grid cell and simulate streamflow at the outlet of

each the watershed.   VIC has been successfully used to simulate major hydrological and thermal processes at different spatial and temporal scales, such as basin streamflow  [*Abdulla and Lettenmaier*, 1997a; b; *Abdulla et al.*, 1996; *Bennett et al.*, 2012; *Bennett et al.*, 2015; *Schnorbus et al.*, 2011], evaporation and transpiration, canopy interception, soil moisture [*Andreadis et al.*, 2005; *Billah and Goodall*, 2012; *Meng and Quiring*, 2008; *Robock et al.*, 2003; *Slater et al.*, 2007; *Wu et al.*, 2015; *Wu et*

*al.*, 2007], surface and ground heat fluxes, ground temperature, and snowpack energy-balance, including ablation and accumulation processes [*Andreadis et al.*, 2009; *Cherkauer and Lettenmaier*, 1999; *Cherkauer et al.*, 2003].

In this study, a fully coupled water-energy balance mode of the VIC model, with 1/64$^{th}$ degree grid resolution (approximately 1km), is used in the LowP and HighP sub-basins of the CPCRW.  We assume

such a higher model resolution in order to explicitly represent and simulate the contrasting hydrological responses of the valley bottoms and hill slopes that exist in a very short spatial distance. The frozen soil algorithm is activated for permafrost grid cells [*Bowling et al.*, 2008; *Cherkauer and Lettenmaier*, 1999; *Cherkauer et al.*, 2003] in order to solve the ground heat flux and surface energy balance. The frozen soil algorithm solves the thermal/heat flux equation through the soil column to calculate the frozen soil

penetration and the ice content of each soil layer [*Cherkauer and Lettenmaier*, 1999]. Frozen soil

algorithm also solves the effect of frozen soil on the moisture transport [*Cherkauer and Lettenmaier*, 1999].

VIC is forced with daily or sub-daily minimum and maximum air temperature, precipitation, and wind speed. The remaining forcing data including incoming shortwave radiation, longwave radiation, atmospheric pressure, and relative humidity are estimated by based on the daily temperature range and precipitation MTCLIM [*Thornton and Running*, 1999] as implemented within the VIC model [*Bohn et al.*, 2013]. The radiation estimate by the VIC model indicates an average of 15-20 $wm^{-2}$ higher shortwave radiation in the LowP sub-basin than the HighP sub-basin. In this study, we generate the daily gridded minimum and maximum temperature, precipitation, and wind speed data from four meteorological stations located within the CPCRW (http://www.lter.uaf.edu/data.cfm **, accessed on May, 2/2013)** using inverse distance weighting (IDW) interpolation method.

In addition to forcing, VIC requires parameters that describe soil properties, vegetation distribution and characteristics, and topographic information.  In this study, a three-layer soil column is used with a top layer with a depth of 0.1 m and variable depth for the second and third layer based on model calibration. Soil properties are uniform within a grid cell, but these properties are allowed to vary for each layer in the grid cell. VIC requires several soil hydraulic and thermal property parameters. Due to a lack of high spatial resolution soil data in the region, most of the soil data — such as soil textural classes, saturated hydraulic conductivity, bulk density and porosity — were extracted directly from the 5 arc minute (approximately  9 km)  Food and Agriculture Organization digital soil map of the world [*FAO*, 1998]. Other soil hydraulic properties —  including field capacity wilting point, residual soil moisture content,

water retention and bubbling pressure — are estimated by the Brooks and Corey formulation [*Rawls and Brakensiek*, 1985; *Saxton et al.*, 1986] based on soil texture classes, porosity and bulk density information. All soil parameters are re-gridded to 1/64$^{th}$ degree. Selected soil property values are shown in Table 2.

VIC uses a mosaic representation of vegetation coverage and subdivides each grid cell's land cover into a specified number of "tiles". Each tile represents the fraction of the cell covered by a particular land cover type (coniferous, evergreen forest, grassland, etc). Vegetation cover composition data used in this study is obtained from the University of Alaska Fairbanks (UAF), Scenarios Network for Alaska and Arctic Planning (SNAP) 1 km X 1 km Land Cover map originating from the North American Land

Change Monitoring System (NALCMS) 2005 data set (http://www.snap.uaf.edu/data.php, accessed on June 23, 2013). Rooting depth data is extracted from The International Satellite Land-Surface Climatology Project (ISLSCP) Initiative II Ecosystem Rooting Depths [*Schenk and Jackson*, 2009], accessed on June 24, 2013). Tree height, trunk ratio, displacement and roughness of the primary vegetation classes are derived from field measurements by *Young-Robertson et al.* [2016]. The

remaining vegetation parameter values including Leaf Area Index (LAI) for each vegetation class in the region are derived from *Hansen et al.* [2000]; [*Myneni et al.*, 1997; *Nijssen et al.*, 2001a; *Nijssen et al.*, 2001b].

## 2.4 The VIC Model Parameterization Schemes

Based on sources of vegetation cover and soil hydraulic property data, we developed two

parameterization schemes: large-scale parameterization (large-scale, hereafter) scheme and small-scale

(small-scale, hereafter) parameterization scheme. In these two schemes, the key differences are in vegetation cover, saturated hydraulic conductivity, and organic layer thickness. The remaining model inputs and parameters including meteorological forcing, vegetation characteristics, and soil properties are the same in both schemes.

### 2.4.1   Large-scale parameterization

The large-scale parameterization scheme is derived from the coarse resolution SNAP vegetation cover (Figure 3d) and FAO soil property (Figure 4a) datasets as described in the previous section (Table 4).

### 2.4.2   Small-scale parameterization

The small-scale parameterization scheme consists of two small-scale parameterization sub-schemes: one derived from fine resolution landscape model (aspect parameterization, hereafter), and one derived from a permafrost map (permafrost parameterization, hereafter).In both small-scale parameterization schemes, the top layer of the permafrost cell is also assumed to be the organic layer [*Ping et al.*, 2005; *Zhang et al.*, 2010] while mineral soil, as obtained from FAO dataset, is assumed for permafrost-free grid cell.

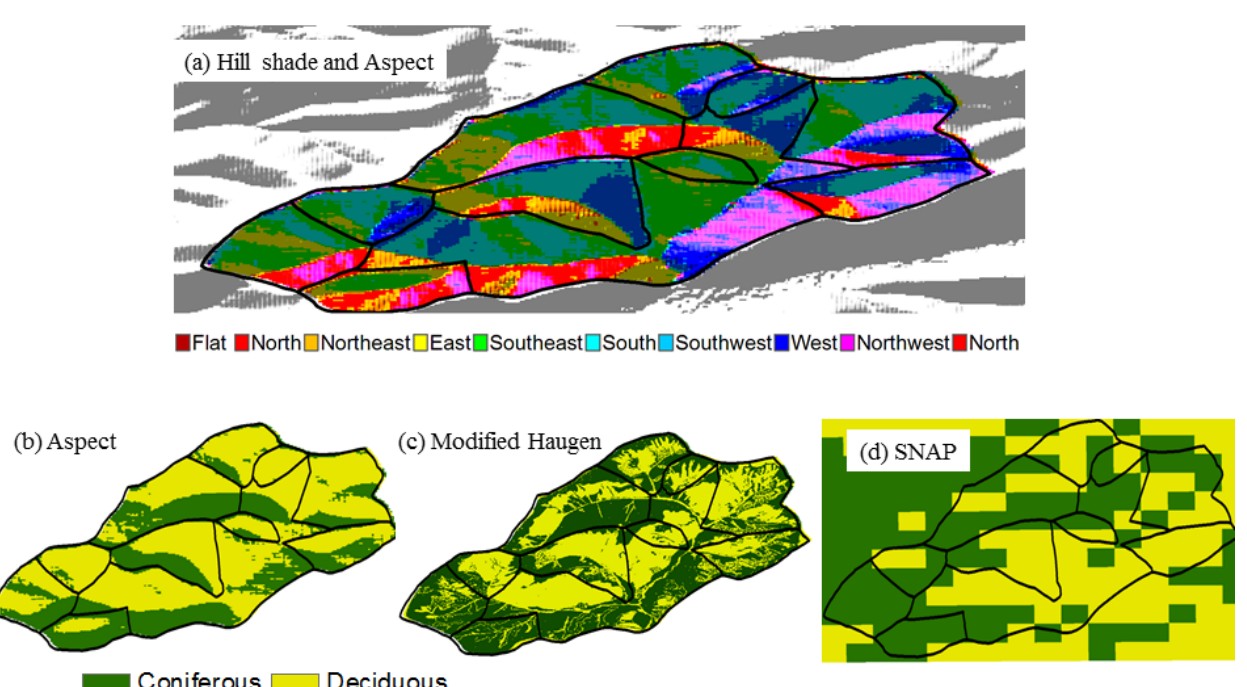

**Figure 3** (a) Aspect map – derived from Digital Elevation Model (DEM) – of the Caribou Poker Creek Watershed (CPCRW). Coniferous and deciduous vegetation composition at the CPCRW as derived from: (b) model-based aspect or topography of the watershed, (c) observation-based vegetation coverage map modified from *Haugen et al.* [1982], (d) large-scale based Scenario Network for Alaska and Arctic Planning (SNAP) vegetation coverage map. Note that (b) is prepared from the aspect map (a) and the fine resolution landscape model.

2.4.2.1 Aspect parameterization

The aspect parameterization is based vegetation cover (Figure 3b) and soil property (Figure 4c) products of the fine resolution landscape model output (Table 4). Saturated hydraulic conductivity and organic layer thickness in the aspect parameterization depends on whether the grid cell is permafrost-underlain or permafrost-free (Figure 4c).



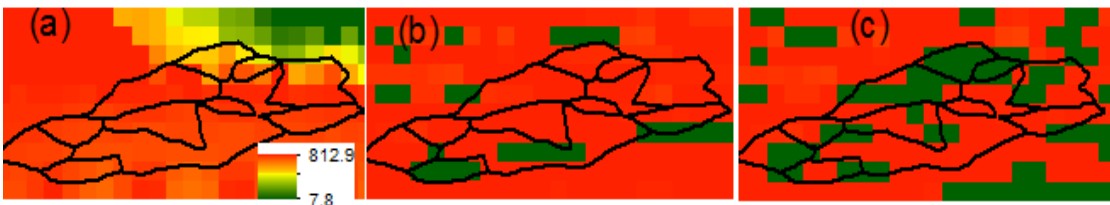

**Figure 4** Saturated hydrologic conductivity (mm/day) as derived from (a) large-scale FAO soil dataset, (b) permafrost map of *Rieger et al.* [1972], and (c) aspect map of CPCRW. Panels (b) and (c) are small-scale parameterizations obtained from altering large-scale values according to the presence or absence of permafrost. Grid cells that are classified as permafrost soil are assigned hydraulic conductivity two-orders of magnitude less than the value obtained from the large-scale (FAO) dataset.

### 2.4.2.2 Permafrost parameterization

The permafrost parameterization is primarily based on the small-scale observed [*Rieger et al.*, 1972] and modeled [*Yoshikawa et al.*, 2002] permafrost maps. In the permafrost parameterization, we assume the proportion of coniferous and deciduous vegetation in each grid cell is the same as the fraction of permafrost-underlain and permafrost-free soil respectively (Table 4). Like the aspect parameterization, we classified a grid cell as permafrost containing when the fraction of permafrost in the grid cell is greater than 0.5 (Figure 4b).

**Table 2** Selected vegetation and soil parameters, their values and estimating methods or sources: runoff influence (*RI*) – runoff increases when parameter values increase (+) and runoff decreases when parameter values increase (-), Coniferous vegetation *(conif),* deciduous vegetation *(decid)*, permafrost-underlain soil *(PF)*, permafrost-free soil *(NPF)*, parameters values specifically for the LowP sub-basin *(LowP)*, parameters values specifically for the HighP sub-basin *(HighP)*.

| Parameters | Value ranges | *RI* | Sources |
|---|---|---|---|
| **Vegetation parameters** | | | |





| Leaf Area Index: $LAI$ | 3.8 – 4.2 (conif) & 0.11 – 6.0 (decid) | (-) | [Hansen et al., 2000; Nijssen et al., 2001a; Nijssen et al., 2001b] |
|---|---|---|---|
| Roughness length: $Z_0$ | 1.23 (conif) & 2.24 (decid) | | |
| Displacement length: $d_0$ | 6.7 (conif) & 13.4 (decid) | | |
| Minimum stomata resistance: $r_{min}$ (s/m) | 130 (conif) & 150 (decid) | | |
| Architectural resistance: $r_{arc}$ (s/m) | 60 (conif) & 60 (decid) | | |
| Canopy albedo: $\alpha$ (fraction) | 0.11 (conif) & 0.09-0.13 (decid) | | |
| Trunk ratio: (fraction) | 0.1 (conif) & 0.4 (decid) | | This study |
| Depth of the two root zone: $r\_depth_1$, $r\_depth_2$ (m) | 0.1, 0.5 (conif) & 0.1, 1.0 (decid) | | Schenk and Jackson [2009] |
| Fraction of root in the two root zone: $r\_fract_1$, $r\_fract_2$ (fraction) | 0.5 , 0.5 (conif & decid) | | |
| **Soil parameters** | | | |
| Porosity: $\rho$ ($m^3/m^3$) | 43.9 – 66.5 | | FAO [1998] |
| Bulk density: ($kg/m^3$) | ~1200 (PF & NPF) | | |
| Exponent used in the estimation of unsaturated hydraulic conductivity: $expt$ | ~ 10.84 (LowP & HighP) | | |
| Exponent used in baseflow curve: $c$ | 2 (LowP HighP) | | |
| Saturated hydraulic conductivity: $K_s$ (mm/day) | ~8 (PF) & ~800 (NPF) | | FAO [1998]; this study |
| Dumping depth: (m) | 4 (LowP) & 2 (HighP) | | |
| **Calibration parameters** | | | |





| Variable infiltration curve parameter: $b$ | 0.11 (*LowP*),  0.31 (*HighP*) | (+) | This study |
|---|---|---|---|
| Maximum velocity of the baseflow: $D_{smax}$ *(mm/day)* | 2.16 (LowP), 2.86 (HighP) | (-) | |
| Fraction of baseflow, $D_{smax}$, where non-linear baseflow begins: $D_s$ *(fraction)* | 0.17 (LowP), 0.45 (HighP) | (-) | |
| Fraction of maximum soil moisture where non-linear baseflow begins: $W_s$ *(fraction)* | 0.79 (LowP), 0.74 (HighP) | (-) | |
| Three soil layer thickness: $d_1$, $d_2$, *and $d_3$ (m)* | 0.1, 0.37, and 0.75 (LowP), and 0.1, 0.35, and 0.47 (HighP) | (-) | |

## 2.5 Calibration and validation

Model calibration is performed by comparing streamflow simulation with observation for the period of 2001 – 2005 at LowP and HighP sub-basin outlets of the CPCRW. The large-scale parameterization scheme – FAO soil (Figure 4a) and SNAP vegetation cover (Figure 3d) datasets – is implemented during the calibration process. The Multi-Objective Complex Evolution (MOCOM) automated calibration approach developed by *Yapo et al.* [1998] was used to match the simulated and observed streamflow. MOCOM is a multi-criteria calibration approach based on random parameter sampling strategy to optimize several user-defined criteria [Wagener et al., 2001; Yapo et al., 1998]. As suggested by *Liang et al.* [1994], the model was calibrated using baseflow generation parameters including: maximum velocity of the baseflow (*Dsmax*), infiltration parameter (*bi*), fraction of *Dsmax* where non-



linear baseflow begins (*Ds*), maximum soil moisture for non-linear baseflow to occur (*Ws*), thickness of the second soil layer (*D2*) and thickness of the third soil layer (*D3*). Table 2 shows final calibration values of the user-defined parameters for both sub-basins.

After calibration parameters were determined, validation of the model with the large-scale and small-scale parameterization schemes were conducted by comparing the observed and simulated runoff at the outlets of LowP and HighP sub-basin for 2005 to 2008. Calibration results obtained from large-scale parameterization is directly implemented into small-scale parameterization during validation in order to examine how process changes with the new parameterizations without re-calibration. We utilized two verification statistics, correlation coefficient ($R^2$, equation 1) and Nash-Sutcliff efficiency (NSE, equation 2).

$$R^2 = \left( \frac{N\left(\sum_{i=1}^{N} S_i . Q_i\right) - \left(\sum_{i=1}^{N} S_i\right).\left(\sum_{i=1}^{N} Q_i\right)}{\left[\left(N\sum_{i=1}^{N} S_i^{\,2} - \left(\sum_{i=1}^{N} S_i\right)^2\right)\left(N\sum_{i=1}^{N} Q_i^{\,2} - \left(\sum_{i=1}^{N} Q_i\right)^2\right)\right]^{0.5}} \right)^2 \qquad (1)$$

$$NSE = 1 - \left( \frac{\sum_{i=1}^{N} (S_i - Q_i)^2}{\sum_{i=1}^{N} (\overline{Q} - Q_i)^2} \right) \qquad (2)$$

where $N$ is equal to the number of data points (i.e. daily streamflow realizations), $i$ is the time step (days), S is the simulated streamflow (mm/day), and $Q$ is the observed streamflow (mm/day).



$R^2$ (equation 1) describes the degree of linear correlation of simulated and observed runoff or goodness of fit. The value of $R^2$ ranges from 0.0 to 1.0, and larger values indicate better fit between simulation and observation. NSE (equation 2) describes the relative magnitude of simulated runoff variances compared to variance in observed streamflow. NSE is also an indicator of model-fit in terms of a scatter

plot of simulated versus observed streamflow values, wherein a slope near the 1:1 line indicates a better fit. The value of NSE ranges from 1 (perfect fit) to $-\infty$. Values between 1.0 and 0.0 are widely considered to be acceptable levels of model performance [*Krause et al.*, 2005]. NSE of below zero indicates that the mean observed streamflow is better predictor than the simulated runoff [*Krause et al.*, 2005].

Percent difference (PD) is also used to indicate and visualize differences in simulated runoff from each parameterization scenarios:

$$PD = \frac{(Q_R - Q_i)}{0.5(Q_R + Q_i)} \square 100\%$$
(3)

Where PD is the percent difference, $Q_R$ is reference observed streamflow and $Q_i$ is simulated runoff. The range of PD (equation 3) falls between -100% to 100% with zero being the perfect match between

observed and simulated streamflow. Large negative PD in the time series indicates model over-prediction. On the other hand, large positive PD represents model under-prediction. PD values closer to zero indicate a better model fit.

### 3.   RESULTS



### 3.1 Comparisons of vegetation cover in each parameterization scenario

Figure 3b, c, and d show the three vegetation cover representations of CPCRW derived from fine

resolution landscape model (aspect), permafrost, and SNAP, respectively. The comparison of vegetation

cover focuses on the coniferous and deciduous vegetation composition in LowP and HighP sub-basins.

5   Table 3 summarizes the proportion of coniferous and deciduous vegetation in the different

parameterization scenarios. The small-scale observed vegetation cover from *Haugen et al.* [1982]

[Figure 3c] is used as a reference vegetation cover to evaluate the other vegetation cover scenarios.

**Table 3** Estimated percent of the sub-basins (LowP and HighP) covered by coniferous and deciduous
vegetation, and percent of the sub-basins underlain by permafrost. Values were obtained by applying
10   different parameterization methods: SNAP vegetation cover (*SNAP*), a modified *Haugen et al.* [1982]
vegetation cover map (*modified Haugen*), aspect-derived vegetation cover and permafrost maps
(*aspect*), the CPCRW permafrost map (*permafrost*), and large-scale FOA digital soil map of the world
(*FAO soil dataset*). *NA* indicates that a given method was not utilized to obtain values for either the
vegetation or permafrost distributions.

| parameterization method | vegetation, % landscape distribution | | | | permafrost, % landscape distribution | |
|---|---|---|---|---|---|---|
| | coniferous | | deciduous | | | |
| | LowP | HighP | LowP | HighP | LowP | HighP |
| SNAP | 63 | 93.1 | 37 | 6.9 | *NA* | *NA* |
| modified Haugen | 30 | 95 | 70 | 5 | *NA* | *NA* |
| aspect | 13.2 | 78 | 86.8 | 22 | 9.5 | 42.3 |
| permafrost map | 3 | 53 | 97 | 47 | 3.4 | 83 |
| FOA soil dataset | *NA* | *NA* | *NA* | *NA* | 0 | 0 |

The SNAP vegetation cover map represents the HighP sub-basin well but overestimates the coniferous





proportion of LowP sub-basin. It also shifts the dominant vegetation type from deciduous to coniferous

for LowP sub-basin (30% in small-scale to 63% from SNAP, see Table 3).  Vegetation cover from fine

resolution landscape model (aspect) captures the dominate vegetation type for both sub-basins (Table 3).

Coniferous and deciduous composition in the permafrost map parameterization is generally similar to

permafrost-underlain and permafrost-free proportion. A 3/97 and 53/47 proportion of

coniferous/deciduous vegetation is obtained from parameterization based on permafrost map in LowP

and HighP sub-basins respectively (Table 3)

### 3.2 Calibration

Figure 5a and b show daily simulated vs. observed streamflow for the calibration period of 2001-2005

in LowP and HighP sub-basins. Table 5 summarizes the coefficient of determination ($R^2$) and Nash-

Sutcliff efficiency (NSE) during the calibration of two sub-basins. Calibration results show that runoff

simulation in HighP sub-basin (Figure 5b) is in better agreement than LowP sub-basin with the

observed streamflow (Figure 5a). Although the $R^2$ for LowP (0.51) is greater than HighP (0.48), the

NSE of LowP is less than HighP (0.17 for LowP and 0.38 for HighP sub-basins) indicating the peak

flows are systematically underestimated in LowP than HighP sub-basin (Table 5).  The simulated

streamflow for both sub-basins is believed to be satisfactory given the inadequate soil and vegetation

parameter representation from the coarse resolution datasets. Moreover, streamflow measurement

during early snowmelt period is mostly missing or not accurate most of the time due to the icing

problem [*Bolton*, 2006] that contributes to the lower calibration values in this region. Figure 5c shows

the observed streamflow differences between LowP and HighP sub-basins for the calibration period of



2001-2005. Streamflow in the LowP sub-basin has lower peak flow, higher baseflow, and longer runoff response to precipitation and snowmelt than HighP. Peak flow from HighP is an order of magnitude greater than the LowP sub-basin. Final calibration parameter values suggest that HighP sub-basin requires parameters that favor direct runoff while LowP sub-basin requires parameters that favor more

5   baseflow and infiltration (Table 2).



**Figure 5** Observed versus simulated streamflow during the calibration period of 2001-2005: (a) at the LowP sub-basins, and (b) at the HighP sub-basin, of the CPCRW. (c) shows the difference in observed streamflow between LowP and HighP sub-basins.

5    **3.3 Hydrological fluxes under different parameterization schemes**

In order to test the improvement of VIC's simulations in small-scale parameterization over the coarser resolution data products, three simulations were completed: one using large-scale parameterization and the other two using small-scale parameterizations (aspect and permafrost). Streamflow, ET, and soil

10  moisture are simulated in the LowP and HighP sub-basins from 2006 to 2008. Table 4 summarizes how the soil property and vegetation cover are parameterized in the three parameterization scheme implementations.

**Table 4** Definition of the parameterization scenarios with respect to corresponding vegetation cover and soil property parameters

| Scenarios | Vegetation cover | Soil hydraulic property |
|---|---|---|
| Aspect | Small-scale landscape model (Aspect) based vegetation cover map (Figure 3b) | Aspect based soil property (Figure 4c) |
| Permafrost | Permafrost distribution map (Figure 1) Permafrost areas are assumed coniferous while the remaining is deciduous | Soil hydraulic property derived from permafrost distribution map (Figure 4b) |
| Large-scale | SNAP vegetation cover (Figure 3d) | FAO soil dataset(Figure 4a) |

*3.3.1   Streamflow*



The simulated hydrograph changes with vegetation cover and soil properties in both sub-basins. Figure 6a and c show the simulated vs observed streamflow of each parameterization scheme in the LowP and HighP sub-basins of the CPCRW. The percent difference (PD, equation 3) of each simulation from observation is also shown in Figure 6b and d. Table 5 compares the three streamflow simulation

schemes against observation in both sub-basins. Streamflow simulation based on large-scale parameterization yields the lowest performance with a $R^2$ /NSE of 0.34/0.17 and 0.48/0.42 for LowP and HighP sub-basins respectively (Table 5). The mean annual streamflow simulation in both sub-basins is largely underestimated under large-scale parameterization (by 45% to 68 % and 27% to 52% of the total annual runoff in LowP and HighP sub-basins). The underestimation is higher in the LowP

sub-basin. Further, the simulated spring peak flow tends to occur earlier than observed peak for both sub-basins (Figure 6).

**Table 5** Coefficient of determination ($R^2$) and Nash-Sutcliff efficiency (NSE) values for the LowP and HighP sub-basins for calibration and validation periods.

|  | LowP Sub-Basin | | HighP Sub-basin | |
|---|---|---|---|---|
|  | $R^2$ | NS | $R^2$ | NS |
| **Calibration** | 0.51 | 0.17 | 0.48 | 0.38 |
| **Validation** |  |  |  |  |
| Large scale parameterization | 0.34 | 0.17 | 0.48 | 0.42 |
| Permafrost Parameterization | 0.43 | 0.3 | 0.51 | 0.48 |
| Aspect Parameterization | 0.51 | 0.48 | 0.53 | 0.52 |

The simulated streamflow hydrograph obtained from aspect parameterization – the small-scale parameterization scheme which is primarily derived from the fine resolution landscape model – is



closest to the observed streamflow hydrograph for both sub-basins (Figure 6a and c) with an $R^2$/NSE of 0.51/0.48 and 0.53/0.52 for LowP and HighP sub-basins respectively (Table 5). Throughout the simulation period, the PD is also close to zero under aspect parameterization than the other two parameterizations (Figure 6b and d). Both peak and low flows in both sub-basins are well simulated in small-scale parameterization scheme (Figure 6). Mean annual PD values are also close to zero for streamflow simulations using aspect parameterization (-7.6% to -14.4% and -2.2% to -8.3% in LowP and HighP sub-basins) compared to simulation under large-scale parameterization (-17.7% to -25.4% and -11.5% to -15.9% in LowP and HighP sub-basins.).

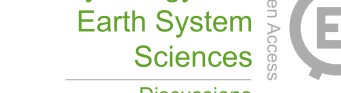

**Figure 6** Comparison of streamflow simulations – obtained from different parameterizations – with observation: (a) in the LowP sub-basin, (c) in the HighP sub-basin. (b) and (d) show the percent difference (PD) for each runoff simulation from the observed streamflow in the LowP and HighP sub-basins, respectively.

### 3.3.2 Evapotranspiration

Unlike streamflow simulations, ET simulations generally do not display significant differences between LowP and HighP sub-basins in all parameterizations. Figure 7a and b show comparisons of simulated ET from the three-parameterizations in the LowP and HighP sub-basins. There are slight variations, however, in the peak and low ET rates in the two sub-basins. ET simulation under the aspect parameterization —for example in LowP—predicts low ET most of the time. However, in the HighP sub-basin, low ET is predicted only in the beginning to mid-summer. In the LowP sub-basin, simulated ET does not display significant variation between the permafrost and large-scale parameterizations (Figure 7a). However, in the HighP sub-basin, the difference is greater, especially in the low ET periods (Figure 7b). Generally, ET simulations in the LowP sub-basin are greater than the HighP sub-basin in all parameterizations. Parameterization based on aspect display the lowest ET simulation in both sub-basins.

We also made comparisons of mean annual ET simulations to evaluate the impact of each parameterization on the annual water balance. The difference in simulated mean annual ET between LowP and HighP sub-basins under aspect parameterization (356.5 and 335.2 mm in LowP and HighP sub-basins) is almost half of the values of ET simulation when permafrost (396.6 and 352.2 mm in LowP and HighP sub-basins) and large-scale parameterization (402.8 and 362.8 mm in LowP and HighP sub-basins) are used.





For further comparison between the LowP and HighP sub-basins (Figure 8), ET and streamflow

simulation under aspect parameterization is selected due to its better streamflow simulation

performance (Figure 6). Generally, ET is higher than runoff most of the time in both sub-basins. The

HighP sub-basin, however, displays more runoff than ET during snowmelt, late summer/fall and large

5    storm events (Figure 8b).

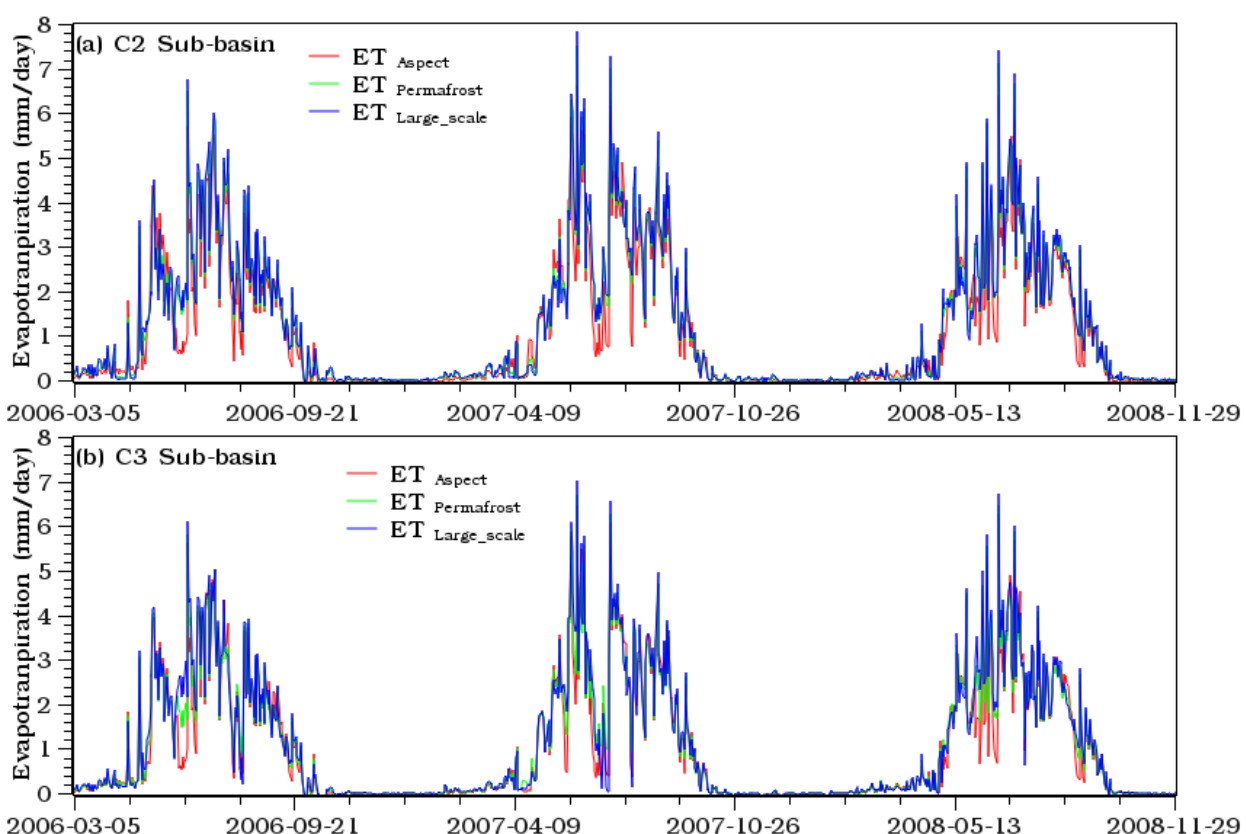

**Figure 7** Areal average evapotranspiration (ET) simulations in each parameterization scenarios: (a) in
the LowP sub-basin and (b) in the HighP sub-basin.



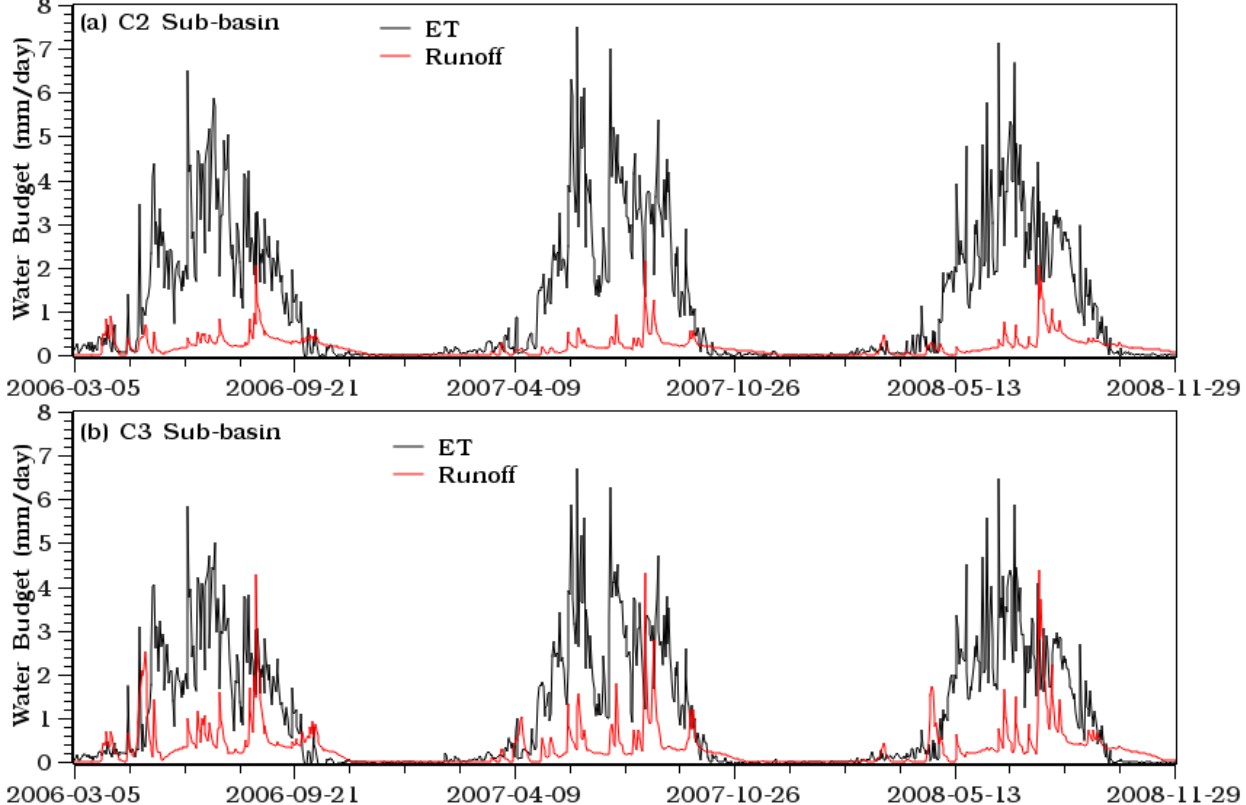

**Figure 8** Comparison of evapotranspiration (ET) and runoff simulations – obtained from parameterization based on aspect: (a) in the LowP sub-basin and (b) in the HighP sub-basin.

### 3.3.3 Soil moisture

Spatial variation of soil moisture simulations further indicates the influence of the spatial representation of vegetation cover and soil property in the model. Figure 9 and Figure 10 show the sub-basin average volumetric soil moisture simulations in the three soil layers in LowP and HighP sub-basins respectively. The results show both frozen and unfrozen soil moisture content. Variation in soil properties and




vegetation cover is generally found to be sensitive to VIC's soil moisture simulation, especially during

spring in the first layer, and throughout the entire simulation period in the second and third soil layer.

**Figure 9** Comparison of watershed integrated oil moisture content simulations between
parameterization scenarios in the LowP sub-basin: (a) in the top soil layer, D1, (b) in the middle soil





layer, D2, and (c) in the bottom soil layer, D3. Note that solid lines show total soil (liquid plus ice) moisture in the soil column and dotted lines show the liquid water portion of soil moisture.

Comparison of the basin integrated frozen and unfrozen soil moisture indicates that the soil columns are thawed during summer for all parameterizations in most part of both sub-basins (Figure 9 and Figure 10). However, this does not mean that permafrost is not simulated anywhere in both sub-basins. The differences in the third layer soil moisture simulations – where permafrost is believed to exist – between LowP and HighP sub-basins clearly show that more areas are permafrost-underlain in the HighP sub-basin than the LowP sub-basin. In both sub-basins, the frozen soil thaws faster – for up to a week – in soil moisture simulations under large-scale and permafrost parameterization schemes than soil moisture simulations under the aspect parameterization scheme. In the thin top soil column, neither the LowP nor the HighP sub-basin displays significant differences in soil moisture content between parameterizations over the summer.

There are, however, some variations in spring, especially during the snowmelt period. Both frozen and unfrozen soil moisture content increase/rise – in the top and middle soil layer – as the snowmelt progresses in the first few days. Later, as the average temperature increases to above freezing, the frozen soil starts to thaw and release moisture, showing a significant decline (in frozen soil) in the first two soil layers of both sub-basins. Thus, the unfrozen soil moisture increases continuously until all the snow has melted. There is no significant variation in these processes between LowP and HighP sub-basins. However, there is significant variation in the rates of snowmelt, melt/refreeze, and soil moisture increase/decrease between the parameterization schemes used. The processes are slightly faster in the LowP sub-basin than in the HighP sub-basin.







**Figure 10** Comparison of watershed integrated oil moisture content simulations between parameterization scenarios in the HighP sub-basin: (a) in the top soil layer, D1, (b) in the middle soil layer, D2, and (c) in the bottom soil layer, D3. Note that solid lines show total soil (liquid plus ice) moisture in the soil column and dotted lines show the liquid water portion of soil moisture.

Top layer soil moisture simulation using the aspect parameterization shows a slower snowmelt and rise in soil moisture early in the spring than the large-scale parameterization. The melt/refreeze process takes longer time in the aspect parameterization as well. Comparison of the time to reach peak soil moisture in the second layer reveals that the aspect parameterization takes a longer time to reach

saturation than the large-scale parameterization. The overall patterns of these rates do not display significant variation between the two sub-basins, although the variability between parameterizations is less defined in HighP sub-basin.

Overall, aspect parameterization results in higher soil moisture in the top and bottom soil layers, and lower soil moisture in the middle layer than the remaining parameterizations in both LowP and HighP

sub-basins. Soil moisture simulation in the LowP sub-basin (Figure 9c) appears to be more sensitive to variations in the soil property and vegetation cover than the HighP sub-basin (Figure 10c) in the bottom soil layer, and vice versa in the middle layer (Figure 9b and Figure 10b).

Unfrozen soil moisture content over the winter does not show any significant variation among parameterizations for both sub-basins (Figure 9and Figure 10). However, LowP tends to have slightly

higher unfrozen soil moisture than HighP in the bottom soil column (Figure 9c and Figure 10c). The HighP sub-basin also freezes faster than the LowP sub-basin in the bottom soil column. A further analysis in Figure 11 indicates that the change in storage in the LowP sub-basin is higher than the HighP sub-basins.



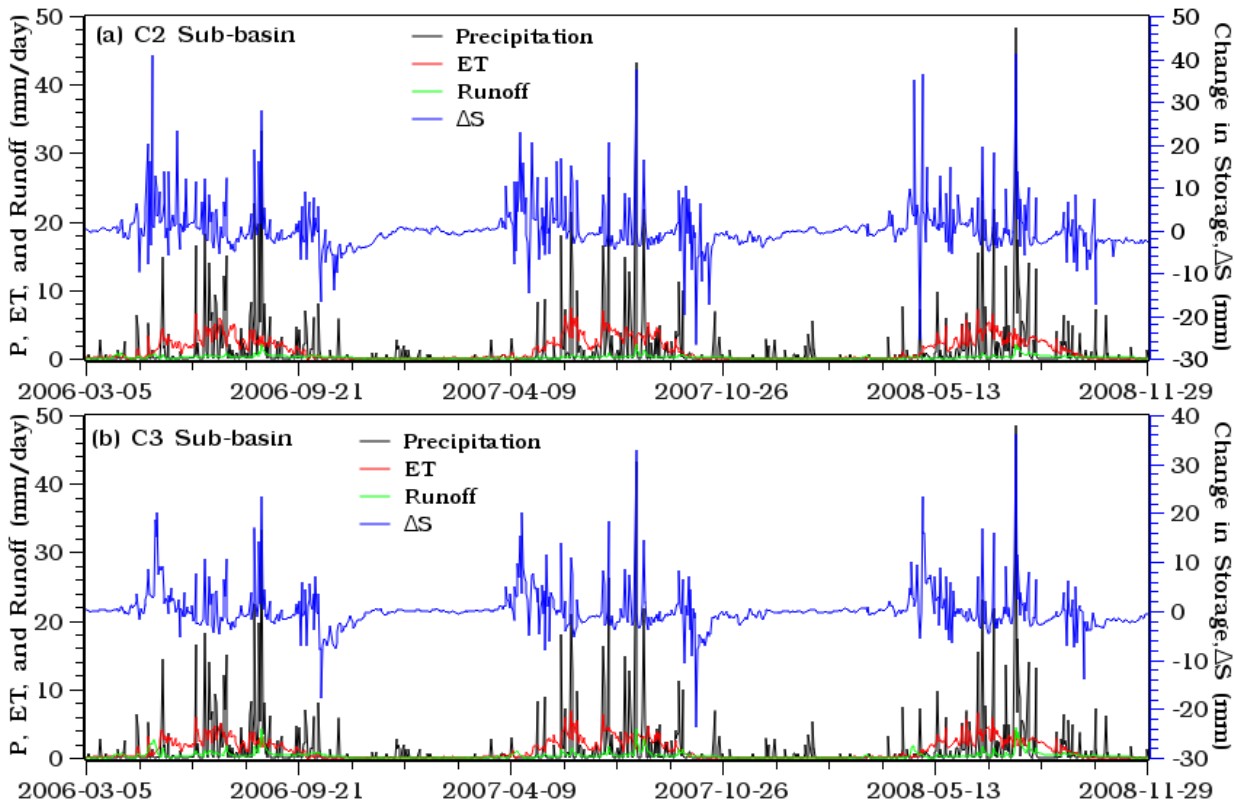

**Figure 11** Partitioning of precipitation or snowmelt (P, Precipitation) into evapotranspiration (ET), runoff, and change in soil moisture storage (ΔS) in the (a) LowP and (b) HighP sub-basins.

5    **4.  DISCUSSION**

One of the important findings from our small-scale landscape model, as shown by Figure 3 and Figure 4, is that an acceptable spatial vegetation cover and soil property representation of the Interior Alaskan sub-arctic can be produced without any direct or remote sensing methods. Large-scale vegetation cover and soil property products that are typically used in meso-scale hydrological modeling do not accurately

10   represent Interior Alaskan boreal forest ecosystem spatial heterogeneities. This is because the coarse





resolution of the data products include regions outside the watershed area that influence the average

value of the grid cell, rather than capturing the spatial heterogeneity of ecosystem properties within the

watershed boundaries. This study indicates that previously documented relationships among soil

hydraulic and thermal properties, vegetation cover, topography, slope and aspect [*Hinzman et al.*, 2006;

*Molders*, 2011; *Morrissey and Strong*, 1986; *Viereck and Van Cleve*, 1984; *Viereck et al.*, 1983] can be

used to formulate a methodology by which local to regional scale landscape properties can be

incorporated into meso-scale hydrological models.

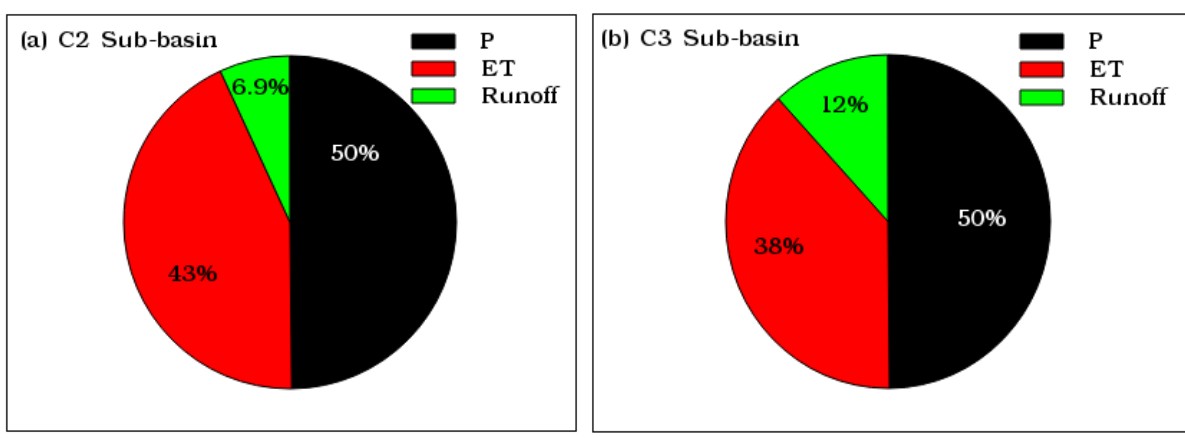

**Figure 12** Mean annual (2006-2008) percentage of water balance components, P-precipitation (black),
ET-evapotranspiration (red), and runoff (green) in the (a) LowP and (b) HighP sub-basins. The change
in storage component is very small, hence not included, compared with the other components.

The soil hydraulic properties obtained from the large-scale FAO soil data do not reflect any variation

between permafrost-affected and permafrost-free soils (Figure 4a). However, many surface and sub-

surface soil hydraulic properties including the hydraulic conductivity and thickness of organic layer

show a large difference between permafrost-affected and permafrost-free soil [*Burt and Williams*, 1976;



*Kane and Stein*, 1983; *Ping et al.*, 2005; *Rieger et al.*, 1972; *Woo*, 1986; *Zhang et al.*, 2009]. Therefore, hydrological modeling using such data creates large uncertainty that cannot be easily corrected with model calibration, as the hydrology and landscape properties including the vegetation composition [*Hinzman et al.*, 2006; *Molders*, 2011; *Morrissey and Strong*, 1986; *Viereck and Van Cleve*, 1984;

*Viereck et al.*, 1983] are primarily controlled by the presence or absence of permafrost [*Bolton*, 2006; *Hinzman et al.*, 2002; *Kane*, 1980; *Kane and Stein*, 1983; *Kane et al.*, 1981; *Petrone et al.*, 2006; *Petrone et al.*, 2007; *Slaughter et al.*, 1983].

Our multi-objective model calibration indicates that watersheds with different permafrost proportions display consistent and contrasting parameter values between nearly permafrost-free and permafrost-

dominated watersheds (Figure 5, Table 2). As reported by previous studies [*Bolton*, 2006; *Bolton et al.*, 2000; *Hinzman et al.*, 2002], simulated runoff in permafrost-free watersheds is best best-fitted with observations for parameter values that favor high infiltration and baseflow and longer recession period (Figure 5a, Table 2). On the other hand, the best-fit between simulated and observed streamflow in the permafrost-dominated watersheds (Figure 5b) is achieved only when baseflow parameters that favor

more direct runoff and inhibit infiltration (Table 2) are introduced into the model. During model calibration, especially for the heavily parameterized distributed models, it is important to consider these variabilities in the baseflow parameters of the discontinuous permafrost watersheds. It could save considerable amount of time and resources, especially for big watersheds and limited computational facilities.

Based on the three model evaluation criteria (equations 1-3), streamflow simulations in both the permafrost-free, LowP, (Figure 6a and b) and permafrost-dominated, HighP (Figure 6c and d) perform

better when the small-scale soil hydraulic and thermal properties (Figure 4c) and vegetation cover (Figure 3b) heterogeneity are introduced into the VIC model's soil property and vegetation cover parameterizations. Comparison of the improvement between the LowP and HighP sub-basins indicates that the strongest streamflow simulation improvement (Table 3and Table 5) was observed in sub-basins

that are highly unrepresented by the large-scale data products – the LowP sub-basin (Figure 3,Figure 4, and Figure 6a and b). In general, our study shows that most of the peak and low flows in both sub-basins are captured well with the small-scale parameterization scheme as compared to the direct use of coarse resolution land surface data products (Figure 6, Table 5), except the 2007 spring flooding event (Figure 6). The exception for the 2007 spring runoff peak can be explained by the high rainfall and

mixed precipitation in the beginning of the 2006/2007 winter (nadp.isws.illinois.edu/data/sites /siteDetails.aspx?id=AK01&net=NTN, accessed May 20, 2013). Rainfall at the onset of winter generally freezes on the surface soil layer over the winter and prevents any snowmelt from infiltrating during spring snowmelt period to generate flooding.

Although we did not detect significant differences in the ET simulations between parameterization

schemes, we found that sub-basin average ET in LowP is higher than HighP, except at the beginning of snowmelt where deciduous trees of the LowP are storing snowmelt water but not yet transpiring [*Young-Robertson et al.*, 2016] (Figure 7 and Figure 8). Unlike other snow-dominated regions in middle latitudes, where ET tends to have strong relationship with precipitation and a change in storage [*Lohmann et al.*, 1998a; b], ET in this region is more strongly related to temperature than precipitation

or change in soil moisture storage. This apparent decoupling between ET and precipitation in the LowP ecosystems is likely because the deciduous trees are utilizing the snowmelt water stored in their trunks

rather than being directly tied to rainfall [*Young-Robertson et al.*, 2016]. This implies that temperature is the limiting factor for ET in this region because the region is not soil moisture limited. This may have important implications for the general warming trend on a broader spectrum, as in the case of permafrost degradation [*O'Donnell et al.*, 2009; *Romanovsky and Osterkamp*, 1995; 2000; *Romanovsky et al.*, 2002].

Several studies have documented that VIC soil moisture simulation is strongly sensitive to how vegetation cover [*Ford and Quiring*, 2013; *Tesemma et al.*, 2015] and soil properties [*Billah and Goodall*, 2012; *Ford and Quiring*, 2013; *Lee et al.*, 2011; *Liang et al.*, 1996; *Lohmann et al.*, 1998b] are represented. This study shows that vegetation cover and soil property representation are not only sensitive to the soil moisture content simulations, but they also have a strong influence in the rate of snowmelt, and snowmelt and refreeze processes in the Alaskan sub-arctic environment. We found that the sensitivity of the soil moisture to soil property and vegetation cover representation is larger in the lower layers than in the top soil layers. In general, given the model being calibrated to fit observed streamflow data, not every process is accurately simulated. However, the small-scale parameterization scheme is the best at capturing the expected soil moisture pattern in both LowP and HighP sub-basins. This includes high soil moisture in the permafrost-dominated sub-basin due to lower hydraulic conductivity of the soil, and the faster winter freeze up of the soil column in the permafrost-dominated sub-basin than the permafrost-free one.

We also made an effort to understand how the annual water balance is partitioned into runoff, ET and change in soil moisture in the permafrost-free, LowP, and permafrost-dominated, HighP, sub-basins. Both sub-basins do not differ in the percentage of each component between the large-scale and small-

scale parameterizations. As shown in the previous studies [*Bolton*, 2006; *Hinzman et al.*, 2002; *Young-Robertson et al.*, 2016], our results indicate that in both sub-basins ET dominants the annual water balances (Figure 8, Figure 11, and Figure 12), while the change in storage is the lowest (Figure 11) in both sub-basins. There are, however, variations in the percentage and pattern of the water balance

components between LowP and HighP sub-basins. The runoff from LowP sub-basin is lower than the HighP sub-basin due to the high tree water storage and transpiration [*Cable et al.*, 2014; *Young-Robertson et al.*, 2016] of the deciduous trees, and higher infiltration of the permafrost-free soil of the LowP sub-basin than HighP sub-basin. The mean annual change in storage in the LowP sub-basin is about 35% higher than the HighP sub-basin. From this result, we can conclude that the soil thermal and

hydraulic properties dictate the partitioning of water into the different processes in this region.

This is the first study that introduces vegetation and soil property heterogeneity from high resolution topographic information into the meso-scale hydrological model. The results need to be verified for regional- scale basins to determine the transferability of our finding to similar areas where the land surface data products do not adequately represent the spatially heterogeneity for accurate hydrological

simulations.

## 5. CONCLUSIONS

This study indicates that coarse resolution soil and vegetation data products — data that are used extensively in land surface modeling in the middle and lower latitudes — do not adequately represent the North American boreal forest discontinuous permafrost ecosystems. Hydrological modeling with

coarse resolution products cannot adequately simulate important small-scale processes. This is because small-scale permafrost distribution and ecosystem composition primarily control the land surface

processes in this region. This indicates the need for landscape modeling that can produce these small-scale features and incorporate them into land surface models. The strong correlative relationship between topography, vegetation, and permafrost distribution [*Burt and Williams*, 1976; *Haugen et al.*, 1982; *Hinzman et al.*, 1998; *Hinzman et al.*, 1991; *Viereck et al.*, 1983] in this region can be used to

produce a fine resolution landscape model that represents the small-scale soil property and vegetation cover heterogeneity for distributed hydrological models.

This study also shows that our fine resolution landscape model – based primarily on this strong relationship between permafrost, topography, and vegetation composition – produced a better representation of permafrost and vegetation cover than large-scale soil and vegetation cover products.

Hydrological simulations — including basin integrated and spatially variable runoff, evapotranspiration and soil moisture dynamics — using a small-scale parameterization scheme derived from a fine resolution landscape model are a significant improvement over parameterizations based on coarse resolution data products.  Finally, in our effort to demonstrate methodologies that can improve hydrological modeling through a small-scale parameterization scheme, we intend to implement and test

results from this pilot study into a large river basin in the next phase of the research.

**Acknowledgments**

This work was supported by the grant from the Department of Energy SciDAC grant # DE-SC0006913.





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
