# Peer review of "Manuscript under review for journal Hydrol. Earth Syst. Sci."

_Hydrology and Earth System Sciences, 2017_

## Referee Comment (RC1) · Anonymous Referee #1 · 5 Apr 2017

The paper introduces a fine-scale parameterization scheme based on a landscape model using 30-m DEM data. The paper futher assesses the hydrological impacts of the fine-scale parameterization scheme compared to a coarse one based on outdated datasets (e.g., FAO soil data and SNAP vegetation cover) in a perfmafrost-dominated 100-km2 catchment. The results show that the streamflow estimates from the small-scale parameterization match the observations slightly better (but still rather poorly), which is kind of obvious. Of course using 9-km FAO data to parameterize a 100-km2 catchment is not going to work and I don't see why anyone would think otherwise. It is also a shame that the approach is tested in only one very small catchment, how much better does the approach work in considerably larger catchments?

[Figure]

In the abstract, it is stated that the two parameterizations "capture most the peak and low flows with similar accuracy in both sub-basins" and then after it states that "on average, the small-scale parameterization improves the total runoff simulation approximately by up to 50% in the LowP sub-basin and 10% in the HighP sub-basin". Which one is true?

On page 13 it is stated that there is a "lack of high spatial resolution soil data in the region". The SoilGrids250m dataset might perhaps be useful (http://journals.plos.org/plosone/article?id=10.1371/journal.pone.0169748).

Page 13: "All soil parameters are regridded to 1/64th degree". How? Nearest neighbour? Bilinear?

So the catchments cover like a single grid cell of the 9-km resolution FAO map. Then how come there is finer-scale spatial variability visible in Figure 4a?

Page 19: Why is only the large-scale parameterization used for the calibration? Wouldn't it be more fair to re-calibrate for both the small- and large-scale parameterizations?

Might be worth mentioning that lumped catchment values are derived using the calibration.

"Values between 1.0 and 0.0 are widely considered to be acceptable levels of model performance". This is not true. Although it depends on the situation, I suppose values >0.5 can generally be considered acceptable.

Figure 6a: The observed streamflow time series look kind of strange, in 2006 in particular the streamflow looks truncated. Could this be due to ice blockage or?

[Figure]

---

## Referee Comment (RC2) · R.A. Woods (Referee) · 8 May 2017

**Toward improved parameterization of a macro-scale hydrologic model in a discontinuous permafrost boreal forest ecosystem**
Endalamaw et al.

Summary

This manuscript develops and tests a method for improved representation of permafrost-related spatial variability of soil and vegetation characteristics in hydrological models. Building on the empirical association between topographic aspect and localised presence/absence of permafrost (and consequential impacts on soils and vegetation), the authors develop two simple, high-resolution predictors of vegetation type and soil properties, resulting in new maps of parameter values for a land surface model. One of the two predictors is a local, site-specific map defining permafrost occurrence, and the other is aspect, the product of topographic analysis using widely available data. They compare their new parameter maps to a more generalised source of this information, using each as input to the VIC model. The authors find that the widely topography-derived information provides a useful improvement over the generalised soil and vegetation data. The result is potentially relevant to any region underlain by discontinuous permafrost whose occurrence is controlled mainly by aspect-driven differences in solar radiation, though the testing is still at a preliminary stage. Although the authors used VIC as their model, it seems that the approach is independent of the model, so may be of quite general interest.

Overall, I found the science question was clearly identified, and the proposed approach sound. To me the results of the testing (better performance measures using the new approach) were encouraging, as opposed to compelling. It's not clear whether the improved representation of soils makes a big improvement to the flow simulations, but it's a step in the right direction. In principle, given the field-based knowledge of permafrost impacts on hydrology, it seems entirely rational to include this kind of permafrost information if it is available.

Main Points

1. Near the end of Section 1, I think there needs to be a short section on strategies for representing subgrid variability in large-scale models.  The general problem is well-known and there are several possible solutions, e.g. mosaic of tiles within a cell, statistical parameterisation, flux-matching parameterisation (MPR method of Samaneigo) ... How does your approach fit into the range of options? Why did you choose your approach?

2. Similarly to the last point, why did you choose VIC? Presumably your approach is quite general, and still applies if the model is not ViC?

3. P5 "by implementing a small-scale parameterization scheme" I found this terminology confusing. The phrase "parameterization scheme" is conventionally used (e.g. in numerical weather models) to describe how the grid-scale flux relates to grid-scale properties, when taking account (implicitly) of sub-grid variability. But in this paper, you are using that term to mean a method of creating grid-scale maps of soil and vegetation parameters. I think the use of the word "scheme" may be misleading to some readers.

4. P17 "we classified a grid cell as permafrost containing when the fraction of permafrost in the grid cell is greater than 0.5" (and also a similar point from P11) What are the implications of doing it this way at the 1 sq km scale? You need to discuss the length scale of the "true" permafrost field vs the model grid scale.

5. Table 3 The HighP catchment shows 42.3% permafrost when derived from aspect, but 83% permafrost when derived from permafrost map (based on a soils map of Rieger 1972). This deserves some comment about how aspect is calculated and up-scaled from 30m to 1km

(Fig 3a). How much of the 1972 permafrost has thawed since? Also, some comment is needed about the authors' decision to rely on the soil map of Rieger to define permafrost, rather than using for HighP the 55% of Table 1, or the 53.2% (and associated map) given by Haugen et al (1982, their Table 1, Figure 3). 83% is very different from 53%! Presumably this decision could have a significant impact if the aspect-based approach was to be scaled up to more of the Alaskan interior?

6. Figs 7-11: I did not find the details of simulated ET and soil moisture very helpful. They revealed what was generally expected from prior knowledge of LowP vs HighP, and no field observations were presented to verify the model outputs (if relevant field observations are available, they would greatly strengthen this part of the paper). The differences between parameterisations were clear once the runoff was shown, provided one knows that long-term storage changes are negligible. Perhaps the figures would be more relevant as supplementary material, whose purpose is to show that the model simulations of ET and soil moisture in these catchments are not inconsistent with process knowledge? With maybe only one key figure retained in the body of the paper?

7. P40 Paragraph starting "Several studies have documented that VIC soil moisture simulation is strongly sensitive to ..." This research question of the sensitivity to soil/veg parameterization of model-simulated ET and soil moisture is difficult to usefully address without observations of those processes. Now that we see VIC is sensitive to the difference in parameter fields, how do we know which parameter field leads to more reliable ET & soil moisture?

8. Usually, models such as VIC are used to assess hydrologic response at scales much larger than a few sq km. Having shown that VIC simulates the difference between LowP and HighP more accurately with the new parameterisation, it would have been interesting to see what difference this makes at larger scales, such as at the outlet of CPCRW. This would also tie in well with your stated aspiration for application on larger domains. Can you include such assessments?

Minor Points

9. P2 "Simulated hydrographs based on the small-scale parameterization capture most of the peak and low flows with similar accuracy in both sub-basins compared to the parameterization based on coarse resolution dataset."  Not clear by what is meant with this last comparison

10. P5 "due to the large differences in the rates of and controls on ET" Do the differences in plant canopy also cause differences in snow interception?

11. P5 "Hydrological modelling using these coarse resolution datasets cannot produce accurate estimates of the spatially variable and basin-integrated watershed responses." You need to say why this cannot be done. I presume the main reason is that the coarse datasets contain spatially-smoothed characteristics, but hydrologic response is not a linear function of the true heterogeneous characteristics?

12. Fig 2: It would be good to mention briefly in the text what is the spatial density of the source data which underlies the gridded climate maps.

13. P10 "In addition, the relationship between vegetation, permafrost, slope and aspect [Hinzman et al., 2006; Morrissey and Strong, 1986; Viereck et al., 1983] is also introduced during the modeling activity." It is not clear what you mean by modelling activity - can you be more specific?

14. P13 During the description of VIC data sources (esp vegetation), it would be useful to remind the reader that these are coarse-scale inputs.

15. P17 When you introduce Fig 4a, I think it is useful to point out to the reader that the FAO soils map does not include any of the permafrost soils which are known to be present in CPCRW.
16. P21 "Values between 1.0 and 0.0 are widely considered to be acceptable levels of model performance [Krause et al., 2005]." I think that higher values of NSE than 0.0 are generally required to reach the "acceptable" standard. Please look a little more widely at the literature to establish a benchmark. In strongly seasonal cold climates, one can achieve values much larger than zero with models which are very poor indeed (See Bettina Schaefli's paper on "Do Nash values have value?").
17. Fig 5 (and later figures): More helpful to refer to LowP and HighP rather than C2 and C3 basins
18. Fig 6: For C-2 the observed streamflow is almost constant for extended periods, and drops suddenly in all 3 years. Is this the way the watershed functions or is it a measurement artifact?
19. Fig 12 - need to explicitly state that this water balance is based only on observations? (is that correct?)
20. P41 "The mean annual change in storage in the LowP sub-basin is about 35% higher than the HighP sub-basin." No evidence was shown to support this point. In Fig 12 it was stated that storage changes are negligible.

---

## Author Comment (AC1) · 10 Jul 2017

**The authors' responses are shown in red text below or next to each reviewer's comments.**

We thank Referee #2 for the comments; below we give the reply to the comments

Summary

This manuscript develops and tests a method for improved representation of permafrost-related spatial variability of soil and vegetation characteristics in hydrological models. Building on the empirical association between topographic aspect and localised presence/absence of permafrost (and consequential impacts on soils and vegetation), the authors develop two simple, highresolution predictors of vegetation type and soil properties, resulting in new maps of parameter values for a land surface model. One of the two predictors is a local, site-specific map defining permafrost occurrence, and the other is aspect, the product of topographic analysis using widely available data. They compare their new parameter maps to a more generalised source of this information, using each as input to the VIC model. The authors find that the widely topography derived information provides a useful improvement over the generalised soil and vegetation data. The result is potentially relevant to any region underlain by discontinuous permafrost whose occurrence is controlled mainly by aspect-driven differences in solar radiation, though the testing is still at a preliminary stage. Although the authors used VIC as their model, it seems that the approach is independent of the model, so may be of quite general interest.

Overall, I found the science question was clearly identified, and the proposed approach sound. To me the results of the testing (better performance measures using the new approach) were encouraging, as opposed to compelling. It's not clear whether the improved representation of soils makes a big improvement to the flow simulations, but it's a step in the right direction. In principle, given the field-based knowledge of permafrost impacts on hydrology, it seems entirely rational to include this kind of permafrost information if it is available.

> We completely agree with the reviewer. Our approach is independent of a specific hydrological model, VIC model in this case. It can be applied to any distributed hydrological models. Our method can be applied to discontinuous permafrost regions in the sub-Arctic, as mentioned by the reviewer. This study will be extended to a larger and regional scale catchment in the Interior Alaska to test the extent to which this approach is valid.

Main Points

COMMENT # 1.

Near the end of Section 1, I think there needs to be a short section on strategies for representing subgrid variability in large-scale models. The general problem is well-known and there are several possible solutions, e.g. mosaic of tiles within a cell, statistical parameterisation, flux-matching parameterisation (MPR method of Samaneigo) ... How does your approach fit into the range of options? Why did you choose your approach?

We thank the reviewer for this suggestion. We strongly agree with comment on including a section that describes how to address the overall problem of representing the sub-grid variability of soil and vegetation cover representation in large scale models, particularly in the Interior Alaska Boreal forest.

ADRESSED in the revised manuscript, the following new paragraph is added (PAGE 6, LINE 3-13).

"Several techniques, including parameter regionalization (Parajka et al., 2005; Pokhrel et al., 2008; Samaniego et al., 2010; Kumar et al., 2013) are proposed to address the limitation of large-scale hydrological simulation due to sub-grid variabilities of soil and vegetation cover properties. Standard regionalization (SR) (Pokhrel et al., 2008; Troy et al., 2008) and multiscale parameter regionalization (MPR) (Samaniego et al., 2010; Kumar et al., 2013) are widely used to parameterize the basin hydrological predictors. The later approach (MPR) takes account sub-grid heterogeneity of hydrological processes by implementing the prior predictors (the relationships between elevation, slope, vegetation characteristics, soil properties and etc) into model parameterization (Samaniego et al., 2010; Kumar et al., 2013). However, the first approach (SR) lacks explicit representation of the basin characteristics and primarily depend on calibration of sub-sets of the region (Troy et al., 2008). In this study, we proposed, a landscape modeling approach of soil property vegetation characteristics parameterization of a mesoscale hydrological model. The landscape modeling approach proposed in this study

closely fits the MPR approach. In both approaches, basin hydrological predictors are explicitly taken into account, and parameters are transferable to scales and locations other than a particular area of similar basin characteristics".

COMMENT # 2.

Similarly to the last point, why did you choose VIC? Presumably your approach is quite general, and still applies if the model is not ViC?

The approach can be applied to any distributed hydrological models as indicated. We selected the VIC model for its mosaic vegetation cover representation, grid based calculation of the water and energy balance by which we can explicitly represent surface heterogeneity by making the grid cell very fine depending on landscape characteristics.

COMMENT # 3.

P5 "by implementing a small-scale parameterization scheme" I found this terminology confusing. The phrase "parameterization scheme" is conventionally used (e.g. in numerical weather models) to describe how the grid-scale flux relates to grid-scale properties, when taking account (implicitly) of sub-grid variability. But in this paper, you are using that term to mean a method of creating grid-scale maps of soil and vegetation parameters. I think the use of the word "scheme" may be misleading to some readers.

We agree with the reviewer that the word "parameterization scheme" could be confusing, especially for atmospheric and climate science expertise. Hence, the word "Parametrization scheme" is changed to "parameterization method" throughout the manuscript in the revised version.

COMMENT # 4.

P17 "we classified a grid cell as permafrost containing when the fraction of permafrost in the grid cell is greater than 0.5" (and also a similar point from P11) What are the implications of doing it this way at the 1 sq km scale? You need to discuss the length scale of the "true" permafrost field vs the model grid scale.

In the model, soil properties are unique for a grid cell, unlike the vegetation cover representation (mosaic representation). A grid cell will have either a permafrost property or permafrost-free property. As can be seen Figure 1 and 4, the permafrost field is in the order of meters. In this case, we need a threshold by which we can assume a 100% permafrost grid cell. After doing several tests, we set 0.5 to be a threshold fraction. If the permafrost extent in a grid cell is more that 50%, we assume 100% permafrost condition otherwise permafrost free.

It is addressed in the revised manuscript.

COMMENT # 5.

Table 3 The HighP catchment shows 42.3% permafrost when derived from aspect, but 83% permafrost when derived from permafrost map (based on a soils map of Rieger 1972). This deserves some comment about how aspect is calculated and up-scaled from 30m to 1km (Fig 3a). How much of the 1972 permafrost has thawed since? Also, some comment is needed about the authors' decision to rely on the soil map of Rieger to define permafrost, rather than using for HighP the 55% of Table 1, or the 53.2% (and associated map) given by Haugen et al (1982, their Table 1, Figure 3). 83% is very different from 53%! Presumably this decision could have a significant impact if the aspect-based approach was to be scaled up to more of the Alaskan interior?

The 55% permafrost in the HighP sub-basin, based on a soils map of Rieger 1972, in table 1 and throughout the manuscript is corrected to 53%. Not more than 1 % of the permafrost is thawed since Rieger 1972 did the measurement a couple decades ago (Yoshikawa et al., 2002). Therefore we are convinced that using that old but small-scale observed permafrost map is a better approach to compare our aspect based permafrost map.

The 83% permafrost proportion is due to the threshold 0.5 value. A grid cell is either 100% permafrost or 100% permafrost free. Most of the grid cells in HighP sub-basin are more that 50% permafrost that made a higher (83%) permafrost condition in the model.

As indicated in section 2.2, the nine aspect classifications are grouped into two categories; Permafrost aspects and permafrost-free aspects. North, northeast, northwest, southwest, and flat aspects are classified as permafrost-underlain aspects. South, southeast, east, and west are classified as permafrost-free aspects. The 30m aspect map is then up-scaled to a 1km grid cell by calculating the fraction of permafrost and permafrost free aspects in each grid cell. This finally gives us the percentage of coniferous and deciduous vegetation, and permafrost extent in each 1km grid cell.

COMMENT # 6.

Figs 7-11: I did not find the details of simulated ET and soil moisture very helpful. They revealed what was generally expected from prior knowledge of LowP vs HighP, and no field observations were presented to verify the model outputs (if relevant field observations are available, they would greatly strengthen this part of the paper). The differences between parameterisations were clear once the runoff was shown, provided one knows that longterm storage changes are negligible. Perhaps the figures would be more relevant as supplementary material, whose purpose is to show that the model simulations of ET and soil moisture in these catchments are not inconsistent with process knowledge? With maybe only one key figure retained in the body of the paper?

We agree with the reviewer that if observed ET and soil moisture were available, the story from figures 7-11 would have been stronger. While we appreciate the reviewer's suggestion on putting them as supplementary materials, we are convinced to keep them in the result sections. Because, there are several points discussed from each figure. Although, runoff plots show the major differences in each parameterization, presenting ET and soil moisture simulations highlight things we need to consider as a next step towards improving our understanding of the water and energy balance. Comparing results based only on runoff may not be sufficient if we can show how the other water and energy balance components behave.

COMMENT # 7.

P40 Paragraph starting "Several studies have documented that VIC soil moisture simulation is strongly sensitive to ..." This research question of the sensitivity to soil/veg parameterization of

model-simulated ET and soil moisture is difficult to usefully address without observations of those processes. Now that we see VIC is sensitive to the difference in parameter fields, how do we know which parameter field leads to more reliable ET & soil moisture?

We agree with the reviewer that sensitivity of ET and soil moisture simulations would have been fully addressed if these observations exist. Although basin scale observation is unavailable, several observation and experimental studies have documented the general soil moisture and ET responses in the permafrost-underlain and permafrost-free environment in the Interior Alaska (Kane & Stein, 1983; Hinzman et al., 1998; Cable & Bolton, 2012; Cable et al., 2014; Young-Robertson et al., 2016; Peckham et al., 2017). Compared to simulations based on the large-scale parameterization, ET and soil moisture simulations based on small-scale parameterization method are consistent with our knowledge from those previous studies.

COMMENT # 8.

Usually, models such as VIC are used to assess hydrologic response at scales much larger than a few sq km. Having shown that VIC simulates the difference between LowP and HighP more accurately with the new parameterisation, it would have been interesting to see what difference this makes at larger scales, such as at the outlet of CPCRW. This would also tie in well with your stated aspiration for application on larger domains. Can you include such assessments?

We completely agree with the reviewer. This study is the first part of the project towards producing an improved mesoscale hydrological modeling in the Interior Alaska. In the next phase, we are expanding the method to a regional scale basin larger than the CPCRW.

Minor Points

COMMENT # 9.

P2 "Simulated hydrographs based on the small-scale parameterization capture most of the peak and low flows with similar accuracy in both sub-basins compared to the parameterization based on coarse resolution dataset." Not clear by what is meant with this last comparison

We addressed this comment in the revised manuscript (PAGE 2, Line 16-18).

CORRECTED TO:

Simulated hydrographs based on the small-scale parameterization capture most of the peak and low flows, with similar accuracy in both sub-basins, compared to simulated hydrographs based on coarse resolution dataset.

COMMENT # 10.

P5 "due to the large differences in the rates of and controls on ET" Do the differences in plant canopy also cause differences in snow interception?

Yes, they do, Included in the revised manuscript (Andreadis et al., 2009) (PAGE 5, LINE 7).

COMMENT # 11.

P5 "Hydrological modelling using these coarse resolution datasets cannot produce accurate estimates of the spatially variable and basin-integrated watershed responses." You need to say why this cannot be done. I presume the main reason is that the coarse datasets contain spatially-smoothed characteristics, but hydrologic response is not a linear function of the true heterogeneous characteristics?

We explained the reason in the revised manuscript as follow (PAGE 5, LINE 17-19):

Hence, hydrological modeling using these coarse resolution datasets cannot produce accurate estimates of the spatially variable and basin-integrated watershed responses. This is primarily because coarse resolution datasets smoothed the small-scale watershed heterogeneities that control hydrological responses of the sub-Arctic watersheds. Until the small-scale hydrologic processes, soil properties, and vegetation distributions are well

represented, accurate large-scale hydrologic simulation and modeling remains extremely challenging (Walsh et al., 2005).

COMMENT # 12.

Fig 2: It would be good to mention briefly in the text what is the spatial density of the source data which underlies the gridded climate maps.

We explicitly addressed this comment in the revised manuscript (PAGE 10, LINE 3-9).

COMMENT # 13.

P10 "In addition, the relationship between vegetation, permafrost, slope and aspect [Hinzman et al., 2006; Morrissey and Strong, 1986; Viereck et al., 1983] is also introduced during the modeling activity." It is not clear what you mean by modelling activity - can you be more specific?

We addressed this comment in the revised manuscript. Corrected to: "In addition, the relationship between vegetation, permafrost, slope and aspect (Viereck et al., 1983; Morrissey & Strong, 1986; Hinzman et al., 2006) are included in the fine resolution landscape modeling".

COMMENT # 14.

P13 During the description of VIC data sources (esp vegetation), it would be useful to remind the reader that these are coarse-scale inputs.

We addressed this comment in the revised manuscript.

COMMENT # 15.

P17 When you introduce Fig 4a, I think it is useful to point out to the reader that the FAO soils map does not include any of the permafrost soils which are known to be present in CPCRW.

We addressed the comment in the revised manuscript.

COMMENT # 16.

P21 "Values between 1.0 and 0.0 are widely considered to be acceptable levels of model performance [Krause et al., 2005]." I think that higher values of NSE than 0.0 are generally required to reach the "acceptable" standard. Please look a little more widely at the literature to establish a benchmark. In strongly seasonal cold climates, one can achieve values much larger than zero with models which are very poor indeed (See Bettina Schaefli's paper on "Do Nash values have value?").

We addressed this comment in the revised manuscript (PAGE 21, LINE 10-15). It is modified as follow:

"The value of NSE ranges from 1 (perfect fit) to $-\infty$. While values larger than 0.0 can be considered as acceptable levels of model performance (Krause et al., 2005; Schaefli & Gupta, 2007), values approaching 1.0 are more preferred depending on the study area. NSE uses of the mean observed value as a reference (Schaefli & Gupta, 2007). Hence, factors that affect the mean value observed streamflow will have a stronger effect on the values NSE. In the Interior Alaska, lower value of NSE can be acceptable due to the large uncertainties of mean observed streamflow, which is resulted from aufeis related measurement errors at beginning of snowmelt runoff season (Bolton, 2006). NSE of below zero indicates that the mean observed streamflow is better predictor than the simulated runoff (Krause et al., 2005)."

COMMENT # 17.

Fig 5 (and later figures): More helpful to refer to LowP and HighP rather than C2 and C3 basins

We addressed this comment in the revised manuscript.

COMMENT # 18.

Fig 6: For C-2 the observed streamflow is almost constant for extended periods, and drops suddenly in all 3 years. Is this the way the watershed functions or is it a measurement artifact?

The watershed generally functions as shown in Figure 6. The LowP sub-basin is, especially, very flat compared to the HighP sub-basin due to the higher infiltration of the mineral soils in spring and high transpiration of the deciduous vegetation in summer.

COMMENT # 19.

Fig 12 - need to explicitly state that this water balance is based only on observations? (is that correct?)

Figure 12 is based on VIC simulation, NOT observation. It is explicitly stated in the revised manuscript.

COMMENT # 20.

P41 "The mean annual change in storage in the LowP sub-basin is about 35% higher than the HighP sub-basin." No evidence was shown to support this point. In Fig 12 it was stated that storage changes are negligible.

As shown in Figure 11, the time series change in storage in the LowP sub-basin is much larger compared to the HighP. However, on the annual mean change is storage in both cases is insignificant compared to the other water balance components. The comparison of the mean change in storage between LowP and HighP sub-basins indicate a 35% higher value in LowP sub-basin.

References

Andreadis, K. M., Storck, P., & Lettenmaier, D. P. (2009). Modeling Snow Accumulation and Ablation Processes in Forested Environments. Water Resources Research, 45(5).

Bolton, W. R. (2006). Dynamic Modeling of the Hydrologic Processes in Areas of Discontinuous Permafrost. Ph. D. dissertation, University of Alaska, Fairbanks.

Cable, J. M., & Bolton, W. R. (2012, 2012). Ecohydrology of Interior Alaska Boreal Forest Systems. Paper presented at the 2012 American Geophysical Union Fall Meeting, San Francisco, CA.

Cable, J. M., Ogle, K., Bolton, W. R., Bentley, L. P., Romanovsky, V., Iwata, H., Harazono, Y., & Welker, J. (2014). Permafrost Thaw Affects Boreal Deciduous Plant Transpiration through Increased Soil Water, Deeper Thaw, and Warmer Soils. Ecohydrology, 7(3), 982-997. doi: 10.1002/eco.1423

Hinzman, L. D., Goering, D. J., & Kane, D. L. (1998). A Distributed Thermal Model for Calculating Soil Temperature Profiles and Depth of Thaw in Permafrost Regions. Journal of Geophysical Research: Atmospheres (1984–2012), 103(D22), 28975-28991.

Hinzman, L. D., Viereck, L. A., Adams, P. C., Romanovsky, V. E., & Yoshikawa, K. (2006). Climate and Permafrost Dynamics of the Alaskan Boreal Forest Alaska's Changing Boreal Forest (pp. 39-61): Oxford University Press New York.

Kane, D. L., & Stein, J. (1983). Water Movement into Seasonally Frozen Soils. Water Resources Research, 19(6), 1547-1557.

Krause, P., Boyle, D. P., & Bäse, F. (2005). Comparison of Different Efficiency Criteria for Hydrological Model Assessment. Advances in Geosciences, 5, 89-97.

Kumar, R., Samaniego, L., & Attinger, S. (2013). Implications of Distributed Hydrologic Model Parameterization on Water Fluxes at Multiple Scales and Locations. Water Resources Research, 49(1), 360-379.

Morrissey, L. A., & Strong, L. L. (1986). Mapping Permafrost in the Boreal Forest with Thematic Mapper Satellite Data. Photogrammetric Engineering and Remote Sensing, 52(9), 1513-1520.

Parajka, J., Merz, R., & Blöschl, G. (2005). A Comparison of Regionalisation Methods for Catchment Model Parameters. Hydrology and earth system sciences discussions, 9(3), 157-171.

Peckham, S. D., Stoica, M., Jafarov, E., Endalamaw, A., & Bolton, W. R. (2017). Reproducible, Component-Based Modeling with Topoflow, a Spatial Hydrologic Modeling Toolkit. Earth and Space Science, n/a-n/a. doi: 10.1002/2016ea000237

Pokhrel, P., Gupta, H. V., & Wagener, T. (2008). A Spatial Regularization Approach to Parameter Estimation for a Distributed Watershed Model. Water Resources Research, 44(12).

Samaniego, L., Kumar, R., & Attinger, S. (2010). Multiscale Parameter Regionalization of a Grid-Based Hydrologic Model at the Mesoscale. Water Resources Research, 46(5).

Schaefli, B., & Gupta, H. V. (2007). Do Nash Values Have Value? Hydrological Processes, 21(15), 2075-2080.

Troy, T. J., Wood, E. F., & Sheffield, J. (2008). An Efficient Calibration Method for Continental-Scale Land Surface Modeling. Water Resources Research, 44(9).

Viereck, L. A., Dyrness, C. T., Cleve, K. V., & Foote, M. J. (1983). Vegetation, Soils, and Forest Productivity in Selected Forest Types in Interior Alaska. Canadian Journal of Forest Research, 13(5), 703-720.

Walsh, J. E., Anisimov, O., Hagen, J. O. M., Jakobsson, T., Oerlemans, J., Prowse, T. D., Romanovsky, V., Savelieva, N., Serreze, M., & Shiklomanov, A. (2005). Cryosphere and Hydrology. Arctic climate impact assessment, 183-242.

Yoshikawa, K., Hinzman, L. D., & Gogineni, P. (2002). Ground Temperature and Permafrost Mapping Using an Equivalent Latitude/Elevation Model. Journal of Glaciology and Geocryology, 24(5), 526-532.

Young-Robertson, J. M., Bolton, W. R., Bhatt, U. S., Cristobal, J., & Thoman, R. (2016). Deciduous Trees Are a Large and Overlooked Sink for Snowmelt Water in the Boreal Forest. Sci Rep, 6, 29504. doi: 10.1038/srep29504

---

## Author Comment (AC2) · 10 Jul 2017

**The authors' responses are shown in red text below or next to each reviewer's comments.**

We thank Referee #1 for the comments; below we give the reply to the comments

COMMENT #

The paper introduces a fine-scale parameterization scheme based on a landscape model using 30-m DEM data. The paper futher assesses the hydrological impacts of the fine-scale parameterization scheme compared to a coarse one based on outdated datasets (e.g., FAO soil data and SNAP vegetation cover) in a perfmafrost-dominated 100-km2 catchment. The results show that the streamflow estimates from the smallscale parameterization match the observations slightly better (but still rather poorly), which is kind of obvious. Of course using 9-km FAO data to parameterize a 100-km2 catchment is not going to work and I don't see why anyone would think otherwise. It is also a shame that the approach is tested in only one very small catchment, how much better does the approach work in considerably larger catchments?

We agree with the reviewer on the general comments. This study introduces a method by which small-scale hydrological properties that are related to the presence or absence of permafrost can be represented in a large-scale hydrological model. These small-scale hydrological properties are not well represented in land surface datasets of different spatial resolution, including the recent high resolution soil property dataset by Hengl et al. (2017). The primary reason we used the FAO soil property and the SNAP vegetation cover datasets is the fact that meso-scale hydrological modeling at regional scale basins largely depend on these products. Our end goal is also to improve hydrological modeling of the Interior Alaska using the approach developed in this study. Indeed, there are several finer scale vegetation and soil property data sets. However, all of them do not show the landscape heterogeneity between permafrost and permafrost-free soils, especially the soil property data sets.

We acknowledge this study is conducted in a very small scale experimental watershed. This watershed is the only watershed where basin scale observed permafrost distribution and vegetation cover exist in the region. This allowed us to develop a methodology to reproduce the soil hydraulic properties and vegetation cover map that can be incorporated

into a distributed hydrological model. It would have been better to do the parameterization at a larger basin if the small-scale measurements exist. However, as suggested by reviewer, the methodology will be tested at a reginal scale watershed in the second phase of the project.

COMMENT #

In the abstract, it is stated that the two parameterizations "capture most the peak and low flows with similar accuracy in both sub-basins" and then after it states that "on average, the small-scale parameterization improves the total runoff simulation approximately by up to 50% in the LowP sub-basin and 10% in the HighP sub-basin". Which one is true?

This part indicates the average improvement of runoff simulation in two sub-basins where this study was conducted. The two sub-basins are the permafrost dominated (HighP) and the nearly permafrost-free (LowP) sub-basins. Compared to the large-scale parameterization, the small scale parameterization improves the total runoff simulation by up to 50 % in the LowP sub-basin and by up to 10% in the HighP sub-basin. This implies the improvement is larger in the LowP sub-basin compared to the improvement in the HighP sub-basin.

COMMENT #

On page 13 it is stated that there is a "lack of high spatial resolution soil data in the region". The SoilGrids250m dataset might perhaps be useful (http://journals.plos.org/plosone/article?id=10.1371/journal.pone.0169748 ).

We thank the reviewer for the suggestion. While this dataset has high resolution soil texture information, it still does not show any observed soil hydraulic property variations between permafrost and permafrost-free soils. Our effort is to introduce the soil hydrological properties that are modified by permafrost. In the Interior Alaska, soils with similar texture have different soil hydraulic properties between permafrost underlain and permafrost free soils. In our approach, however, we were able to reproduce the permafrost distribution map from the high resolution DEM data and then represent permafrost soil hydraulic properties in hydrological models.

COMMENT #

Page 13: "All soil parameters are regridded to 1/64th degree". How? Nearest neighbour? Bilinear?

Bilinear interpolation was used to resample the course resolution soil property data to the 1/64th degree model resolution.  It is corrected to "resampled to 1/64$^{th}$ degree using the bilinear interpolation" in the revised manuscript.

COMMENT #

So the catchments cover like a single grid cell of the 9-km resolution FAO map. Then how come there is finer-scale spatial variability visible in Figure 4a?

The map is updated in the revised manuscript. The problem was created during legend creation. However, as pointed out in the comment, the entire basin displays the same soil hydraulic conductivity value in Figure 4a.

COMMENT #

Page 19: Why is only the large-scale parameterization used for the calibration? Wouldn't it be more fair to re-calibrate for both the small- and large-scale parameterizations?

Thank you for raising this point. We had the same feeling. However, in order to see the extent the small-scale parametrization improves hydrological simulation, we prefer not to recalibrate the model again. If we calibrate with the small-scale parameterization, clearly, the lumped parameters will not be the same. So, the improvement could be partly due to calibration. However, if we use the same calibration parameter, any difference made between simulations is 100% due to the variation in the parameterization scenarios. Hence, we preferred not to re-calibrate the model.

COMMENT #

Might be worth mentioning that lumped catchment values are derived using the calibration.

We addressed this comment in the revised manuscript.

"After the lumped sub-basin baseflow generation parameter values are derived by calibration, validation of the model with the large-scale and small-scale parameterization schemes were conducted by comparing the observed and simulated runoff at the outlets of LowP and HighP sub-basin for 2005 to 2008."

COMMENT #

"Values between 1.0 and 0.0 are widely considered to be acceptable levels of model performance". This is not true. Although it depends on the situation, I suppose values >0.5 can generally be considered acceptable.

We agree with the reviewer. Higher values of NSE are generally acceptable depending on the location of the study area. We modified our argument in the revised manuscript as follow.

"While values larger than 0.0 can be considered as acceptable levels of model performance (Krause et al., 2005; Schaefli & Gupta, 2007), values approaching 1.0 are more preferred depending on the study area. NSE uses of the mean observed value as a reference (Schaefli & Gupta, 2007). Hence, factors that affect the mean value observed streamflow will have a stronger effect on the values NSE. In the Interior Alaska, lower value of NSE can be acceptable due to the large uncertainties of mean observed streamflow, which is resulted from aufeis related measurement errors at beginning of snowmelt runoff season (Bolton, 2006). NSE of below zero indicates that the mean observed streamflow is better predictor than the simulated runoff (Krause et al., 2005)".

COMMENT #

Figure 6a: The observed streamflow time series look kind of strange, in 2006 in particular the streamflow looks truncated. Could this be due to ice blockage or?

The streamflow response in most of the permafrost free (LowP sub-basin in this case, Figure 6a) areas is generally flat. Most of the snowmelt and rainfall is infiltrated to the lower soil yarer (Kane, 1980; Kane & Stein, 1983), stored in the tree trunk during snowmelt season (Young-Robertson et al., 2016), and transpired (Cable & Bolton, 2012;

Cable et al., 2014; Young-Robertson et al., 2016). However, in the permafrost-affect soils (HighP sub-basin, Figure 6c), the runoff response is fast and flushy due to the impermeable permafrost layer that blocks water from infiltrating to the lower layer. Hence the strange look of the observed streamflow in Figure 6a is not due to the ice blockage but to the higher infiltration and transpiration loss compared to runoff loss.

References:

Bolton, W. R. (2006). Dynamic Modeling of the Hydrologic Processes in Areas of Discontinuous Permafrost. Ph. D. dissertation, University of Alaska, Fairbanks.

Cable, J. M., & Bolton, W. R. (2012, 2012). Ecohydrology of Interior Alaska Boreal Forest Systems. Paper presented at the 2012 American Geophysical Union Fall Meeting, San Francisco, CA.

Cable, J. M., Ogle, K., Bolton, W. R., Bentley, L. P., Romanovsky, V., Iwata, H., Harazono, Y., & Welker, J. (2014). Permafrost Thaw Affects Boreal Deciduous Plant Transpiration through Increased Soil Water, Deeper Thaw, and Warmer Soils. Ecohydrology, 7(3), 982-997. doi: 10.1002/eco.1423

Hengl, T., De Jesus, J. M., Heuvelink, G. B., Gonzalez, M. R., Kilibarda, M., Blagotić, A., Shangguan, W., Wright, M. N., Geng, X., & Bauer-Marschallinger, B. (2017). Soilgrids250m: Global Gridded Soil Information Based on Machine Learning. PloS one, 12(2), e0169748.

Kane, D. L. (1980). Snowmelt Infiltration into Seasonally Frozen Soils. Cold Regions Science and Technology, 3(2), 153-161.

Kane, D. L., & Stein, J. (1983). Water Movement into Seasonally Frozen Soils. Water Resources Research, 19(6), 1547-1557.

Krause, P., Boyle, D. P., & Bäse, F. (2005). Comparison of Different Efficiency Criteria for Hydrological Model Assessment. Advances in Geosciences, 5, 89-97.

Schaefli, B., & Gupta, H. V. (2007). Do Nash Values Have Value? Hydrological Processes, 21(15), 2075-2080.

Young-Robertson, J. M., Bolton, W. R., Bhatt, U. S., Cristobal, J., & Thoman, R. (2016). Deciduous Trees Are a Large and Overlooked Sink for Snowmelt Water in the Boreal Forest. Sci Rep, 6, 29504. doi: 10.1038/srep29504

---

## Author Comment (AC3) · 10 Jul 2017

We thank both Referees for their detail review and feedback. A letter to the editor and the revised manuscript are attached as a supplement.

Please also note the supplement to this comment: https://www.hydrol-earth-syst-sci-discuss.net/hess-2017-25/hess-2017-25-AC3-supplement.zip

---

## Author Response (AR1)

August 1, 2017

Dear Dr. Van Dijk,

With this letter, we submitted the revised version of our manuscript "Towards improved parameterization of a macro-scale hydrologic model in a discontinuous permafrost boreal forest ecosystem" by Abraham Endalamaw, W. Robert Bolton, Jessica M. Young-Robertson, Don Morton, Larry Hinzman, Bart Nijssen (doi.org/10.5194/hess-2017-25)

We thank you and the two reviewers for the detailed comments and improvement suggestions. We made all the final suggestions changed in the revised manuscript.

As suggested, we changed the 'fine resolution landscape model' to 'sub-grid parameterization method', and 'small-scale parameterization method' back to 'small-scale parameterization scheme'.

In our conclusion, we replace 'significant' by 'notable' as the test was conducted in only two small sub-basins. We also recommend the method to be tested at a larger catchment so that its transferability can be verified.

In the calibration and discussion sections, we added few sentences that discuss how our parameterizations reproduce the runoff contrast between the two catchments.

The discussion of ET and soil moisture is significantly reduced in the revised manuscript. As suggested few ET and soil moisture contrasts between the two catchments are presented.

All the minor comments are also addressed:

1. Page 5, "Hence" ….
   Is corrected to " Hence, hydrological modeling using these coarse resolution datasets cannot produce accurate estimates of the spatially variable and basin-integrated watershed responses. This is primarily because coarse resolution datasets smoothed the non-linear small-scale watershed heterogeneities that control hydrological responses of the sub-Arctic watersheds
2. Page 7, State resolution: The resolution is stated in the revised manuscript (1 km)
3. Fig. 1) Pls indicate in caption how the permafrost was mapped.
   It is addressed by adding 'The permafrost map of CPCRW was produced by a small-scale observation across the watershed (Rieger et al., 1972; Yoshikawa et al., 2002).
4. Fig 5) flow duration curves might also be insightful?
   Flow duration curve is added in Figure 5

Tracked-changes versions of the revised manuscript along the final revision copy are attached as a new author's response comment.
We look forward to hearing from you,

Regards,
Abraham Endalamaw

[revised manuscript text omitted]